# HHU24SWDSCS: A shallow-water depth model over island areas in South China Sea retrieved from Satellite-derived bathymetry

Yihao Wu [1],[*], Hongkai Shi [2],[*], Dongzhen Jia [2], Ole Baltazar Andersen [3], Xiufeng He [2], Zhicai Luo [4], Yu Li [2], Shiyuan Chen [2], Xiaohuan Si [2], Sisu Diao [2], Yihuang Shi [2], Yanglin Chen [2]

[1] School of Geomatics Science and Technology, Nanjing Tech University, Nanjing, 211816, China

[2] School of Earth Sciences and Engineering, Hohai University, Nanjing, 211100, China

[3] DTU Space, Technical University of Denmark, Lyngby, 2800, Denmark

[4] MOE Key Laboratory of Fundamental Physical Quantities Measurement, School of Physics, Huazhong University of Science and Technology, Wuhan, 430074, China

[*] These authors contributed equally to this work.

*Correspondence to*: Hongkai Shi (shk@hhu.edu.cn) and Dongzhen Jia (jdz@hhu.edu.cn)

**Abstract.** Accurate shallow-water depth information for island areas is crucial for maritime safety, resource exploration, ecological conservation, and offshore economic activity. Traditional approaches such as shipborne sounding and airborne bathymetric light detection and ranging (LiDAR) surveys are expensive, time-consuming, and are constrained in politically sensitive regions. Moreover, satellite altimetry-predicted depths exhibit large errors over shallow waters. In contrast, satellite-derived bathymetry (SDB), estimated from multispectral imagery, provides a rapid, open-source, and cost-effective technique to comprehensively characterize the bathymetry of a region. Given the scarcity of in-situ water-depth data for the South China Sea (SCS), a shallow-water depth model, HHU24SWDSCS (Hohai University 2024 Shallow-Water Depth Model of South China Sea), was developed using linear band model by integrating 1298 Ice, Cloud, and land Elevation Satellite (ICESat-2) tracks with 70 Sentinel-2 multispectral images. The model covers over 120 islands and reefs in the SCS region at a resolution of 10 m. Validation against independent ICESat-2 depth data yielded a root mean square error for the model of 0.53-1.24 m (<5% of the maximum depth). Further validation using independent airborne LiDAR bathymetry data in the Lingyang Reef demonstrated an accuracy of 1.01 m. Comparisons with existing bathymetry models revealed the superior performance of the developed model. While traditional bathymetry models exhibit errors up to tens of meters or larger over island regions, and should therefore be used with caution, the HHU24SWDSCS model demonstrated good accuracy in shallow waters across the SCS. This model thus provides a reference for mapping shallow-water depth close to islands and provides fundamental support for research in oceanography, geodesy, and other disciplines. The HHU24SWDSCS data are freely available at https://doi.org/10.5281/zenodo.13852568 (Wu et al., 2024a)

**Key Words.** Shallow water depth, Satellite-derived bathymetry, ICESat-2 photon, Sentinel-2 multispectral image, South China Sea.

**Short summary.** We developed a high-quality and cost-effective shallow-water depth model for >120 islands in the South China Sea, using ICESat-2 and Sentinel-2 satellite data. This model accurately maps water depths with an accuracy of ~1 m. Our findings highlight the limitations of existing global bathymetry models in shallow regions. Our model exhibited superior performance in capturing fine-scale bathymetric features with unprecedented spatial resolution, providing essential data for coastal construction, environmental protection, and marine activities.

# 1. Introduction

Shallow-water bathymetry, which critically important for maritime safety, ecological conservation, and marine economic development (Cesbron et al., 2021; Mavraeidopoulos et al., 2017; Wölfl et al., 2019; Yen et al., 2004), has long been a core research focus in oceanography, geophysics, and coastal geomorphology, profoundly influencing studies on ocean currents, the Earth's gravity field, and seafloor sedimentation processes (Babonneau et al., 2013; Tinto et al., 2019; Wang et al., 2018b; Wu et al., 2024b). Moreover, since most marine-related human activities are concentrated in coastal shallow-water areas, accurate bathymetry information plays a pivotal role in port construction, marine fisheries, cross-sea bridge construction, and other marine economic and engineering activities (Bergstad et al., 2019; Parker, 2002; Šiljeg et al., 2019).

The South China Sea (SCS), one of the most active marine systems globally, is characterized by complex bathymetry (Hwang, 1999; Pitcher et al., 2000; Su et al., 2018). In the central basin of the SCS, the bathymetry is deeper than 4000 m, yet it contains numerous islands, shoals, and banks, with depths <100 m in the continental shelf region (Ruan et al., 2020). A thorough investigation of shallow-water bathymetry in the SCS is crucial for conserving biodiversity, coral reef ecosystems, and marine fisheries, for addressing coastal erosion, and for petroleum exploration; moreover, it is indispensable for achieving sustainable use of marine resources, promoting marine environmental protection, and fostering international cooperation (Folorunso and Li, 2015; Goodman et al., 2020; Misra and Ramakrishnan, 2020; Yen et al., 2004).

Traditional methods for obtaining bathymetry data primarily include shipborne sonar sounding, airborne bathymetric light detection and ranging (LiDAR), and satellite altimetry (An et al., 2024; Guenther, 2007; Smith and Sandwell, 1994). Shipborne sounding, and particularly multibeam sounding, is one of the most accurate methods, capable of simultaneously emitting multiple pulses to expand the survey range and achieve centimeter-level accuracy in water-depth measurements (Costa et al., 2009; Ernstsen et al., 2006). However, shipborne surveys are limited in shallow and narrow waters, in which vessel-draft limitations, beam angles, and multipath effects significantly affect data quality and limit its availability (Costa et al., 2009; Hsu et al., 2021; Schneider von Deimling and Weinrebe, 2014). Airborne bathymetric LiDAR technology can rapidly obtain sub-meter-resolution bathymetric data; however it is costly and its measurement accuracy is influenced by water quality, making it unsuitable for large-scale surveys (Tysiac, 2020). Over deep waters, satellite altimetry-predicted depths play a dominant role in global bathymetry detection (Ge et al., 2025); however, this method faces challenges in coastal zones, and the predicted water depths exhibited large uncertainties in shallow waters (Ferreira et al., 2022). Furthermore, satellite altimetry predicted-depths lack short-wavelength information (i.e., for wavelengths shorter than several kilometers), owing to the limited resolution of altimetry data, thus preventing its use in detecting fine seafloor topography (Wu et al., 2023).

Traditional satellite altimetry-predicted depth and in situ data have been used to develop global bathymetry models, including the SRTM15 series (15″ × 15″) (SRTM: Shuttle Radar Topography Mission) (Tozer et al., 2019) and Topo series of models (1′ × 1′) (https://topex.ucsd.edu/pub/global_topo_1min/) provided by the Scripps Institution of Oceanography (SIO); the DTU series (1′ × 1′) (https://ftp.space.dtu.dk/pub/DTU18/1_MIN/), developed by the Department of Space Research and Technology at Denmark Technical University (DTU Space); and GEBCO (the General Bathymetric Chart of the Oceans) series (15″ × 15″) (https://www.gebco.net/data_and_products/gridded_bathymetry_data/), compiled by the GEBCO Bathymetric Compilation Group. With the accumulation of bathymetric data and advancements in modeling, these models have achieved significant improvements in terms of spatial resolution and accuracy. However, owing to the scarcity of in situ data over shallow waters in the SCS, these models are limited in the accuracy of the bathymetric information, exhibiting data gaps, low spatial resolution, and large uncertainty for shallow water areas (Wu et al., 2023). Consequently, the existing bathymetry models fail to deliver a unified,

high-accuracy representation of the bathymetry in these areas.

In comparison, the Ice, Cloud, and land Elevation Satellite-2 (ICESat-2), equipped with the Advanced Topographic Laser Altimeter System (ATLAS), provides worldwide open-source water depths with an accuracy of 0.43-0.6 m and along-track resolution of 0.7 m (Abdalati et al., 2010; Markus et al., 2017; Martino et al., 2019). Moreover, satellite-derived bathymetry (SDB) technology, utilizing satellite multispectral/hyperspectral imagery, provides comprehensive bathymetric data coverage (Albright and Glennie, 2020; Ma et al., 2020). SDB establishes the relationship between reflectance and water depth, and by combining ICESat-2 data with satellite imagery, SDB can be used to map shallow water bathymetry with an accuracy of ~1 m and resolution of a few meters (Hodúl et al., 2018; Jia et al., 2023; Ma et al., 2020). SDB utilizes openly available data and provides a rapid, accurate, and cost-effective way to capture shallow-water depths with unparalleled accuracy and spatial resolution on a global scale, providing significant advantages over traditional approaches (Ferreira et al., 2022).

Given the lack of accurate water depths near island areas in the SCS, we focused on developing a high-quality shallow-water depth model with a unified spatial resolution using SDB by integrating ICESat-2 data with Sentinel-2 multispectral imagery. The model called HHU24SWDSCS represents a high-quality shallow-water depth (SWD) model over the SCS developed by Hohai University in 2024. This model, which covers >120 islands and reefs in the SCS, is expected to serve as a potential substitute for existing bathymetry models in fields such as oceanography, geodesy, environmental sciences, and marine production activities in the shallow waters over the SCS. The rest of this study is organized as follows: In Section 2, we introduce the study area and data. Section 3 presents the principles for the preprocessing of the ICESat-2 data and for SDB estimation. Section 4 presents the modeling results and examines the model's performance, with validation against independent ICESat-2 and airborne LiDAR data. The performance of the latest global bathymetry models (DTU18BAT, topo_27.1, SRTM15+ V2.6, and GEBCO_2024) is evaluated and analyzed. Section 5 presents the conclusions.

## 2.   Study area and data

The study area was the SCS (Figure 1), with the latest high-resolution bathymetric model (GEBCO_2024, 15″ × 15″) providing the background bathymetry data (GEBCO Bathymetric Compilation Group 2024, 2024). The SCS, a marginal sea in the western Pacific Ocean (3°-22°N, 105°-120°E), is one of the most important maritime passages globally. Located in Southeast Asia, it covers ~3.5 million square kilometers, making it one of the largest and deepest marginal seas (>5000 m deep in the Manila Trench) in the western Pacific Ocean (Wang et al., 2018a; Zhu et al., 2021). Over 100 islands and reefs are scattered across the SCS; these can be geographically divided into four archipelagos: the Xisha Islands, Zhongsha Islands, Dongsha Islands, and Nansha Islands, with the latter accounting for >70% of the islands (Huang et al., 1994). The water depth around these islands and reefs is generally <50 m, and their diameters range from 2 to 25 km (As depicted in Figure 1). Conventional techniques, such as shipborne and airborne surveys, encounter numerous challenges in acquiring shallow-water depths over these islands across the SCS; this is particularly true for the Nansha Islands, where political factors prohibit the use of in situ surveys for water depth meansurements. However, the wide range of the ICESat-2 data and Sentinel-2 imagery provides a solid database for employing SDB to develop a shallow-water depth model covering these islands and reefs (Hsu et al., 2021; Ma et al., 2020). Since most of the Zhongsha Islands comprise submerged shoals for which the ICESat-2 data do not provide valid seafloor topography information, this study focuses on SDB modeling over the Xisha, Dongsha, and Nansha Islands (The geographical locations are displayed in the red boxes in Figure 1(a)). As the Nansha Islands are larger than the other two archipelagos, the Nansha Islands were divided into five subareas for results presentation, resulting in seven subareas in total, as shown in Figure 1. Areas 1 and 2 cover the Xisha and Dongsha Islands (Figure 1(b)

and (c), respectively), and areas 3-7 comprise the Nansha Islands (Figure 1(d)-(h)). It is worth noting that the SDB modeling here is focused on shallow waters near islands and reefs, which is largely affected by the quality of ICESat-2 and Sentinel-2 multispectral data as well as water quality, water condtions and bottom types (Wu et al., 2023). Further information on the subareas is presented in Table 1.

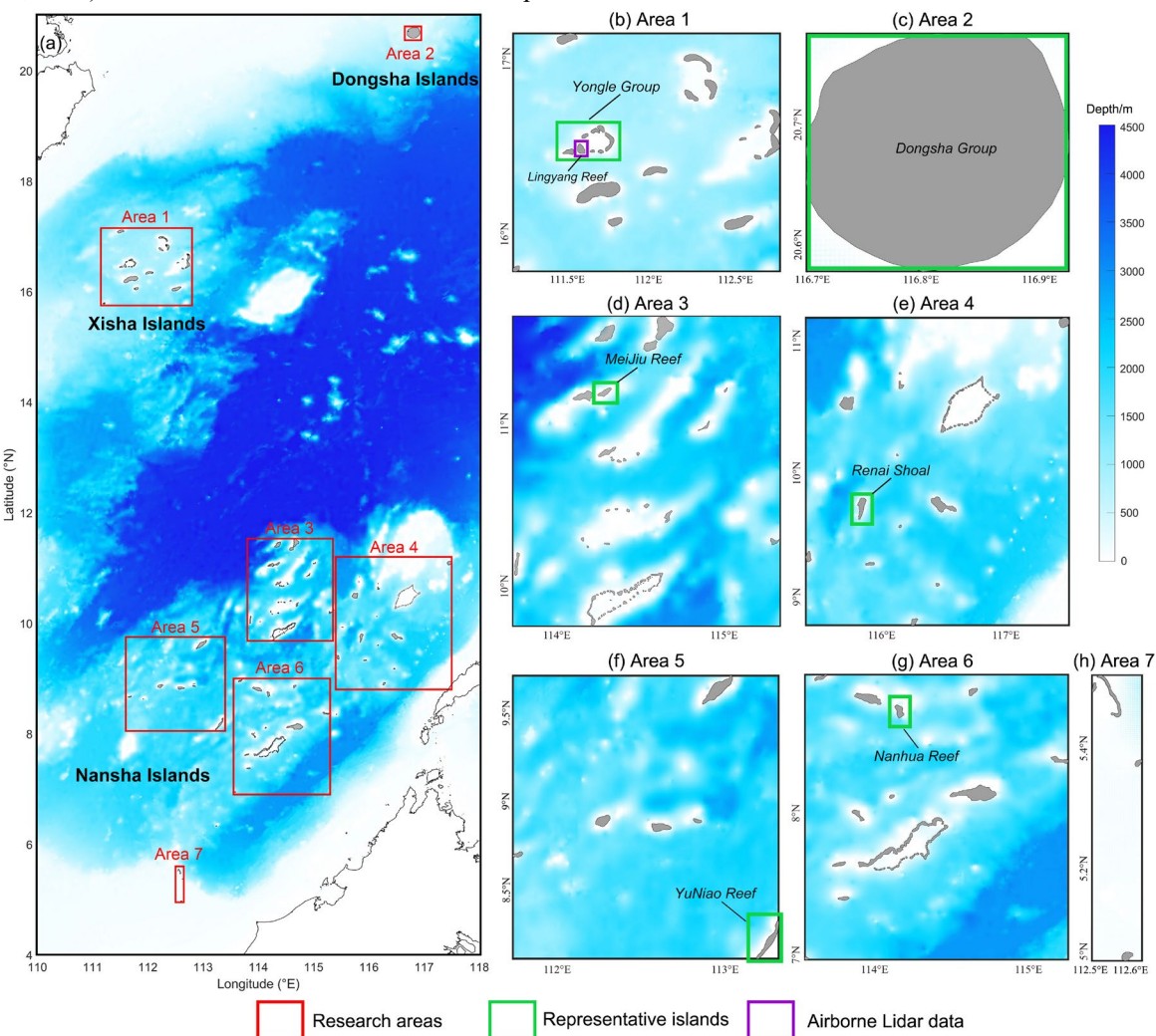

**Figure 1. (a) Distribution of the islands and reefs over the South China Sea (SCS), and seven areas in the red boxes display the subareas. (b)~(h) represent Areas 1~7, respectively. The purple box in (b) displays the location of the airborne LiDAR data in the Lingyang Reef over Xisha Islands. The six green boxes in (b)~(g) show the representative islands. The GEBCO_2024 model is used as background.**

**Table 1. Description of selected subareas in the South China Sea (SCS).**

| Subarea | Latitude (°N) | Longitude (°E) | Number of Islands or reefs |
|---------|---------------|----------------|----------------------------|
| Area 1  | 15.75–17.15   | 111.15–112.80  | 36 |
| Area 2  | 20.55–20.80   | 116.65–116.95  | 2 |
| Area 3  | 9.68–11.53    | 113.80–115.35  | 49 |
| Area 4  | 8.80–11.20    | 115.40–117.50  | 17 |
| Area 5  | 8.05–9.75     | 111.60–113.40  | 6 |
| Area 6  | 6.90–9.00     | 113.55–115.30  | 15 |
| Area 7  | 4.95–5.60     | 112.50–112.65  | 3 |

## 2.1. ICESat-2 data

ICESat-2, launched by NASA in September 2018, has a revisit cycle of ca. 91 days and enables continuous monitoring of changes on the Earth's surface. ICESat-2 is equipped with the latest ATLAS, which emits laser pulses at a 10 kHz pulse-repetition frequency in six beams, achieving an along-track resolution of ~0.7 m and a ranging accuracy better than 1 m (Markus et al., 2017; Martino et al., 2019). It is capable of penetrating water at depths >30 m below the sea surface in clean waters and can measure bathymetry in shallow waters (Guo et al., 2022). For SDB modeling, we utilized the ICESat-2 L2A Global Geolocated Photon Data (ATL03) V006 data-product (https://www.earthdata.nasa.gov/), in which each photon contains information such as latitude, longitude, along-track distance, off-nadir angle, data quality, elevation, and geophysical corrections for factors including solid Earth tides, ocean pole tides, and atmospheric delays (Neumann et al., 2021).

We utilized ICESat-2 data for 2018-2024, encompassing 512 tracks for the Xisha Islands, 73 for the Dongsha Islands, and 1038 tracks for the Nansha Islands, totaling 1623 tracks, with a maximum depth of approximately 30 m. Owing to the difficulty in obtaining in situ water depths (e.g., multibeam sounding data) around islands and reefs, the ICESat-2 data are used for both training and validating the SDB model. The training and validation sample sizes are selected by ensuring that the spatial distribution of both the training and validation data is uniform. This not only allows for better data control but also ensures good spatial coverage for the validation data. The ratios of ICESat-2 data used for the training and validation data are approximately 80% and 20%, respectively. The training (red tracks) and validation data (green tracks) over the seven subareas are illustrated in Figure 2. Notably, ICESat-2 data are acquired independently using individual beams, hence the tracks do not influence one another and are not correlated. This ensures that the training data remains entirely independent from the validation data, allowing for objective assessment of the quality of the computed SDB data.

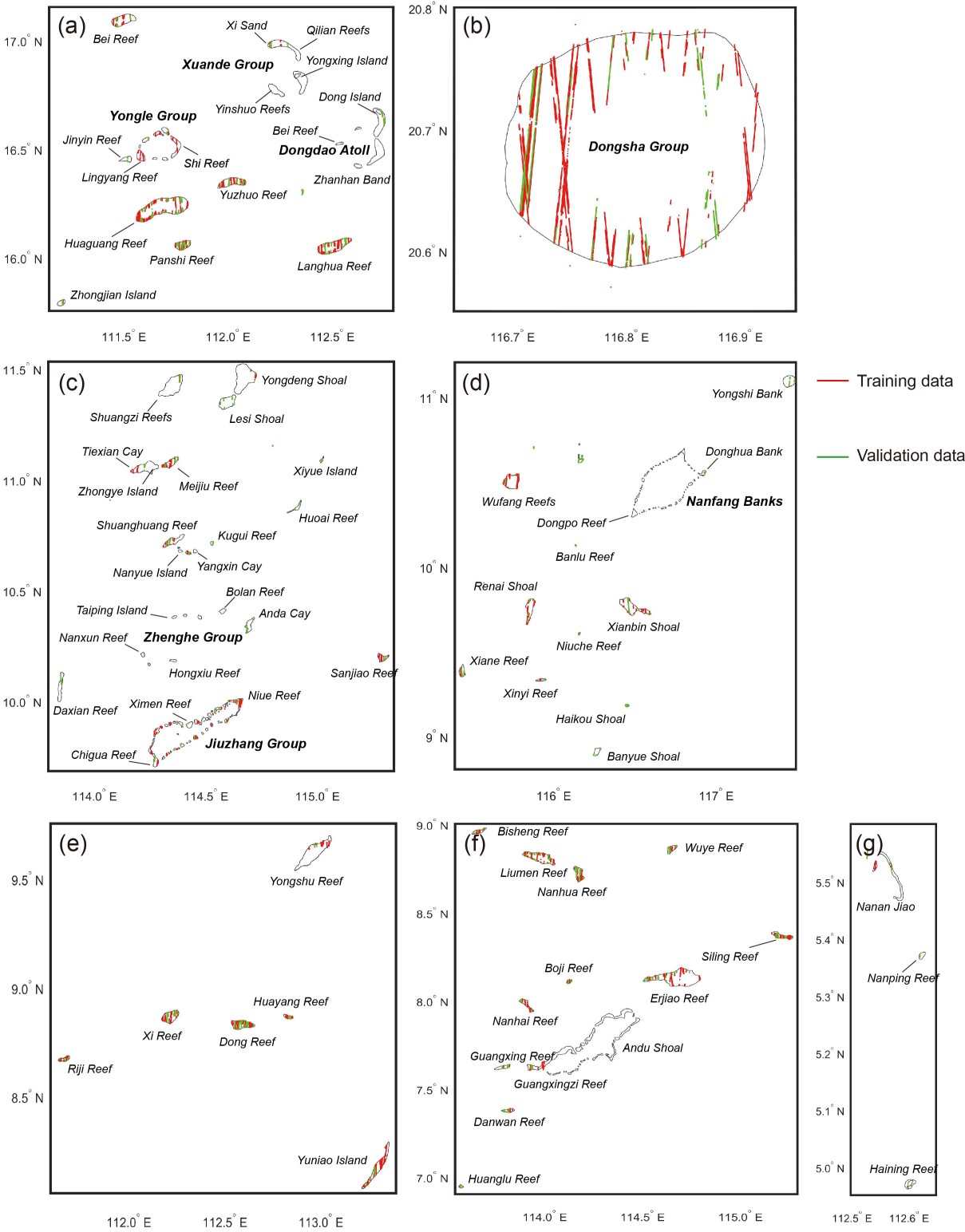

**Figure 2. Distribution of the ICESat-2 training tracks (red) and validation tracks (green) over the seven subareas (a-g, respectively).**

## 2.2.    Sentinel-2 multispectral imagery

High-resolution multispectral imagery from Sentinel-2A and Sentinel-2B Level-2A (L2A) product provided by the European Space Agency (ESA) was utilized for SDB modeling. Sentinel-2A and Sentinel-2B were

launched by the European Space Agency in June 2015 and March 2017, respectively (Drusch et al., 2012). Each carries a Multi-Spectral Instrument and can capture 13 different spectral bands at 443-2190 nm, including blue, green, red, near-infrared, red-edge, and shortwave infrared bands. Their spatial resolution is 10 m, with a swath width of 290 km and a revisit period of 5 days (Gatti and Bertolini, 2015). Owing to its high spatial resolution and short revisit interval, the Sentinel-2 imagery is suitable for SDB modeling. In addition, the L2A products have been atmospherically corrected, offering surface reflectance for SDB modeling.

Here, the spectral information from the red, green, and blue bands was extracted from the Sentinel-2 multiband imagery, based on our preliminary finding that using these three bands yielded better results than using other combinations (Wu et al., 2023). To reduce the effects of temporal changes on bathymetry estimation, only images within the time-span of the ICESat-2 data were selected (Wu et al., 2023, 2025). The AI EARTH platform (engine-aiearth.aliyun.com) was used to select images with minimal cloud cover and sun glint; 70 images were chosen, including five for Xisha Islands, one for Dongsha Islands, and 64 for Nansha Islands. Given that Sentinel-2 multispectral imagery is subject to various environmental interferences, including sun glint, cloud cover, and image noise, coupled with substantial variations in water quality and seafloor composition across different island regions, this study performs separate modeling for each individual island or adjacent islands (e.g., Figure 3(a)). This choice is based on the premise that external influencing factors maintain relative consistency within smaller geographical units, consequently minimizing their potential impact on the accuracy of the SDB results.

To assess the efficacy and applicability of the SDB modeling approach, six representative reefs with diverse geographical distributions, topographical features, and hydrological conditions were selected for presentation (Table 2). Figure 3 depicts the representative islands and reefs, highlighted in the green boxes in Figure 1, including the Yongle Group (Area 1), the Dongsha Group (Area 2), Meijiu Reef (Area 3), Renai Reef (Area 4), Yuniao Reef (Area 5), and Nanhua Reef (Area 6). Figure 3 presents multispectral images (synthesized from the blue, green, and red bands), ICESat-2 water depth, and shallow-water masks (white polygons), along with preselected deep-water areas (purple box). The SDB modeling is conducted within the shallow-water masks of each reef. Notably,GEBCO_2024 model is employed to identify and remove deep-water effects (>100 m) in SDB estimation. The elimination of deep-water effects is intended to reduce sun glint interference and reflection energy from the water body, which in turn helps to establish a more accurate relationship between seafloor reflection energy and water depth (Jia et al., 2023).

To obtain a precise shallow-water mask for a specific island, we first use the normalized difference water index (NDWI) in Sentinel-2 imagery to extract a raw mask, where the green and near-infrared bands are utilized to compute NDWI (Gao, 1996). However, the performance of NDWI in water mask identification is affected by the bottom type and water turbidity, which may lead to misclassification issues (Kirby et al., 2024; Yang et al., 2017b), and we further use the valid ICESat-2 seafloor topography information to correct and refine the initial water mask obtained from the NDWI. Sentinel-2 pixels exhibiting valid bathymetric signals from ICESat-2 were classified as shallow water areas. This refined classification was then incorporated into the water mask, allowing for subsequent data screening and quality control.

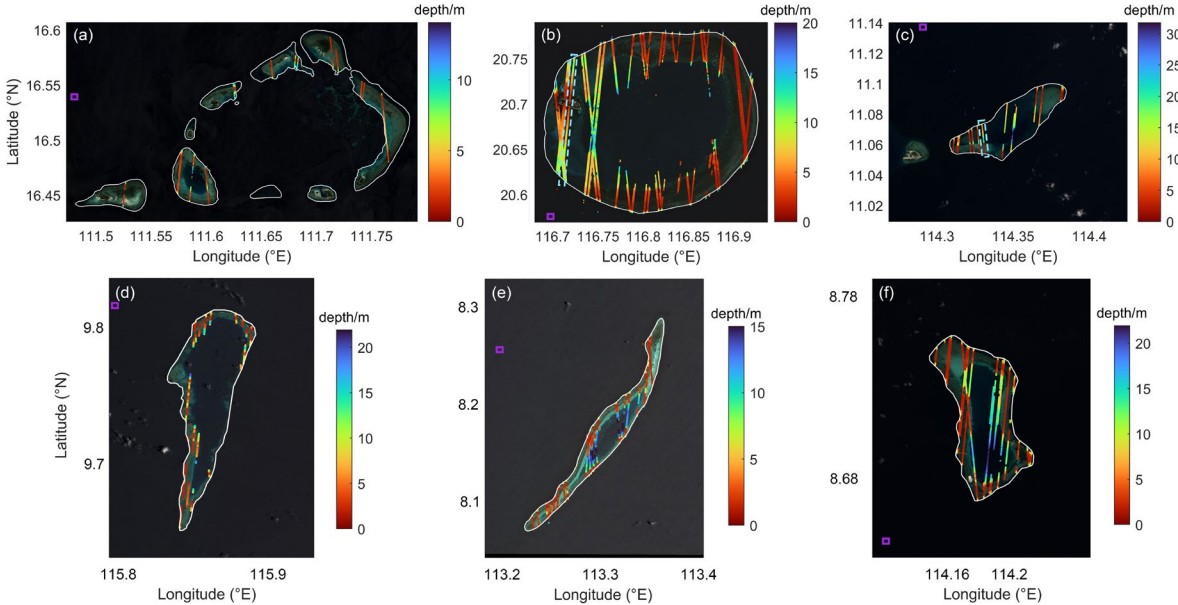

**Figure 3. Representative islands in the research area. (a) Yongle Group, (b) Dongsha Group, (c) Meijiu Reef, (d) Renai Reef, (e) Yuniao Reef, and (f) Nanhua Reef. White polygon denotes the shallow water mask; the tracks indicate the ICESat-2 water-depth data; the purple boxes indicate the reference deep-water areas; and the cyan dotted boxes in (b) and (c) indicate the typical nighttime and daytime ICESat-2 tracks shown in Figures 6 and 7, respectively. The background images are synthetized from the Sentinel-2 red, green and blue band.**

**Table 2. Information of the representative islands**

| Islands name | Latitude (°N) | Longitude (°E) | ICESat-2 tracks | Sentinel-2 image |
|---|---|---|---|---|
| Yongle Group | 16.43–16.60 | 111.48–111.79 | 19 | 20210817T025549 |
| Dongsha Group | 20.55–20.80 | 116.65–116.95 | 73 | 20230207T023901 |
| Meijiu Reef | 11.05–11.10 | 114.30–114.39 | 15 | 20240207T023859 |
| Renai Reef | 9.65–9.8 | 115.83–115.90 | 10 | 20240323T023531 |
| Yuniao Reef | 8.05–8.30 | 113.20–113.37 | 25 | 20200327T024541 |
| Nanhua Reef | 8.66–8.76 | 114.15–114.21 | 26 | 20190228T023631 |

## 2.3. Airborne LiDAR bathymetry

We used airborne LiDAR bathymetric data, provided by the Shanghai Institute of Optics and Fine Mechanics (SIOFM), to independently validate the SDB modeling results (Li et al., 2022; Yang et al., 2022). The airborne LiDAR system (Mapper5000) features a dual-frequency design, including a 1064 nm near-infrared surface channel and a 532 nm green channel for shallow and deep-water detection, with a pulse repetition frequency of 5 kHz. Operated at a flight altitude of 300 to 1100 m and flight speed of 150 to 220 km/h, this system ensures efficient and accurate data collection. The raw data were preprocessed by SIOFM, using procedures including waveform peak detection and range determination, overlapping waveform decomposition, and range-difference correction, based on proprietary algorithms (Yang et al., 2022). The accuracy of the airborne LiDAR water-depth data for Lingyang Reef is ~20 cm (Li et al., 2022). As illustrated in Figure 4(a), the LiDAR data cover the northwestern region of Lingyang Reef in the Yongle Group, with water depths of 0–5 m and an effective point-cloud size exceeding 440,000 points.

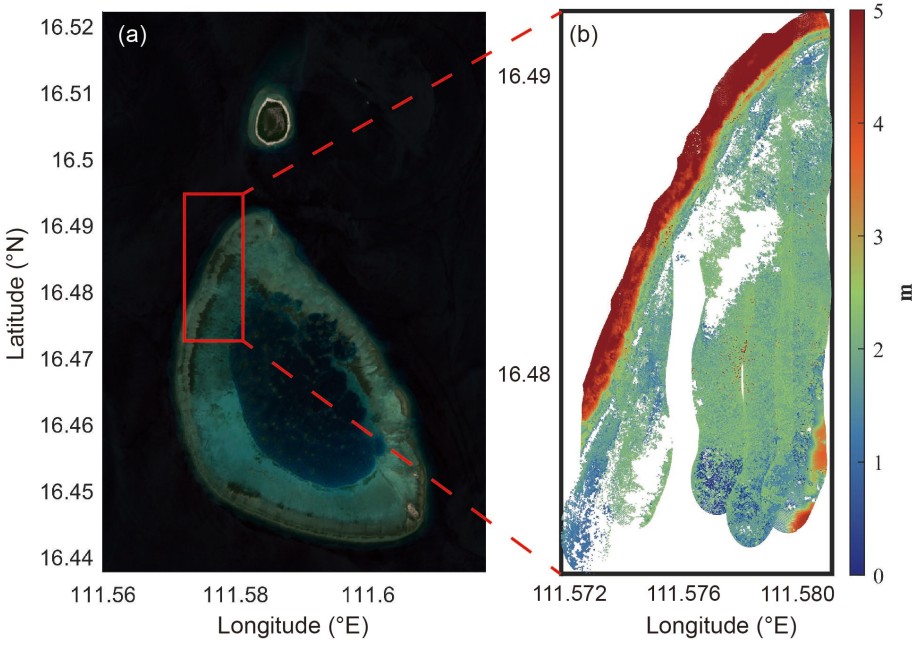

**Figure 4. (a) The Sentinel-2 image of Lingyang Reef and the distribution of the airborne LiDAR data, and (b) the zoom-in view of airborne LiDAR water depth data.**

## 2.4. Global bathymetry models

The recently released bathymetry models, including DTU18BAT (DTU Space, https://ftp.space.dtu.dk/pub/DTU18/1_MIN/), topo_27.1 (https://topex.ucsd.edu/pub/global_topo_1min/), SRTM15+ V2.6 (SIO, https://topex.ucsd.edu/pub/srtm15_plus/), and GEBCO_2024 (GEBCO Bathymetric Compilation Group, https://www.gebco.net/data_and_products/gridded_bathymetry_data/), were introduced for validation and analysis. DTU18BAT was developed by DTU Space in 2019, integrating the GEBCO dataset with satellite altimetric gravity anomalies (Andersen and Knudsen, 2008). It incorporates three years of Sentinel-3A data and seven years of Cryosat-2 observations, offering a spatial resolution of 1′. The topo_27.1 model, released by the SIO, also provides bathymetry at a spatial resolution of 1′ (Smith et al., 2014). Derived from ship soundings and satellite altimetric gravity anomalies, this model continues the tradition of progressively refined seabed topography representations. SRTM15+ V2.6, updated by the SIO, features a finer spatial resolution of 15″ (Tozer et al., 2019). This model combines shipboard soundings with satellite altimetric gravity data to provide an enhanced depiction of seafloor topography, reflecting a comprehensive analysis of available datasets. GEBCO_2024, the recent edition in the GEBCO series, builds upon SRTM15+V2.5.5 as its foundational dataset (Weatherall et al., 2015). Enhanced with in-situ measurements such as echo soundings, seismic records, and lidar, the GEBCO_2024 model offers interpolated satellite altimetry-predicted depths in areas lacking direct measurements. Representing the pinnacle of global seafloor terrain depiction, it embodies an advanced understanding of underwater topography. Notably, these models have been interpolated to align with the SDB models, facilitating accurate comparisons and analyses.

## 3. Methodology

To perform SDB modeling for shallow water regions of a specific island area, ICESat-2 data are first extracted using a shallow-water mask, followed by classification, denoising, refraction correction, and reference datum unification, to obtain high-precision shallow-water depth information. Then, the reflectance information

within the shallow-water mask is extracted from the optimal Sentinel-2 image, and a reference deep-water correction is applied to reduce the sun glint effect. Subsequently, the extracted reflectance information is matched with the processed ICESat-2 depth information, and a Pearson correlation analysis is performed to reduce anomalous data. Based on multivariate linear band model (LBM), regression training is performed using all ICESat-2 depth training data and their corresponding reflectance to solve for the regression parameters. Finally, the SDB estimation is derived based on the reflectance data of the shallow-water area and the LBM.

## 3.1. ICESat-2 data preprocessing

To obtain effective bathymetric data from the ICESat-2 point cloud, this study adopts a sea surface and seafloor identification method based on the point-cloud density distribution (Hsu et al., 2021). This involves noise removal, point-cloud density estimation, sea-surface identification, and water-depth point-cloud extraction. Refraction corrections and reference datum unification are then applied to derive accurate water-depth measurements.

(1) Sea-surface identification

We propose a method for sea-surface identification based on the anisotropic point-cloud density. First, considering the distribution characteristics of the sea surface and bathymetric point data, an elliptical sliding window is constructed to capture their geometric profiles. For each photon $p_i = (x_i, y_i), i = 1, \ldots, N$, an elliptical window is established at $p_i$, and the number of point data within the elliptical window is derived:

$$d_{ij} = \sqrt{\left(\frac{x_j - x_i}{a}\right)^2 + \left(\frac{y_j - y_i}{b}\right)^2}$$

$$D_i = \{j \mid d_{ij} \leq 1, j = 1, \ldots, N, j \neq i\}$$

$$(1)$$

where $a = 50\,m$ and $b = 2\,m$ denote the major and minor axes of the elliptical window, respectively; $d_{ij}$ represents the distance of $p_j$ relative to the ellipse; $D_i$ is the number of points within the elliptical window centered at $p_i$; $N$ is the total number of point data; and $x_i, y_i$ represent the along-track distance and point-cloud elevation, respectively.

For each point, the number of point data within its elliptical neighborhood is computed, normalized, and used as the point-cloud density distribution:

$$\rho(x_i) = \frac{D_i - \min(\mathbf{D})}{\max(\mathbf{D}) - \min(\mathbf{D})}$$

$$(2)$$

where $\rho$ denotes the point-cloud density map and $\mathbf{D}$ represents the vector of the number of neighboring point data for each point.

Subsequently, the statistical analysis based on the Scott's rule is performed to estimate the noise threshold (Scott, 1979), as follows:

$$T_{noise} = \frac{3.5 \cdot \sigma}{\sqrt[3]{N}}$$

$$(3)$$

where $\sigma$ represents the standard deviation of $\rho$. Point data exceeding the noise threshold are removed.

Next, the point cloud density map is discretized into a grid with a cell resolution of 0.5 m in elevation and an along-track resolution of 30 m. This density grid is then stacked in the along-track direction to accumulate the density for each elevation cell. By calculating the gradient of the accumulated density, we identify the elevations corresponding to the maximum and minimum gradient values, thereby locating the boundary of the sea surface point cloud. Consequently, the point data within this boundary are extracted and fitted to estimate sea-surface

height (SSH); this is then used as the instantaneous SSH at the time of the ICESat-2 measurement.

 (2) Bathymetry point-cloud extraction

After removing the detected sea-surface point cloud, the density grid of the remaining point cloud and the noise threshold are recalculated, and point data exceeding 0.5 times the noise threshold are removed. To more accurately extract the bathymetry information, the maximum density points are identified along the depth direction, and the point cloud within ±1 m of the maximum density point is marked as the bathymetry point cloud. Considering the measurement accuracy of ±1 m in the ICESat-2 bathymetry data, a local weighted least squares (LS) fitting algorithm is used to extract the bathymetry.

 (3) Refraction correction

Refraction is one of the most significant factors influencing the accuracy of ICESat-2 laser bathymetry (Yang et al., 2017a), which is deduced by the different propagation speeds of light in different media (such as in air and seawater). In shallow-water areas, the refraction effect is more pronounced, owing to the influence of sea-surface waves and the resulting change in depth. We initially estimated shallow-water depths by computing the difference between the seafloor photon-derived depth and the corresponding sea surface height. However, considering the time difference between the acquired ICESat-2 and Sentinel-2 data used for modeling, a unified reference datum is required as the preliminary depth information. We therefore used the latest DTU22MSS model as the reference datum for bathymetric data correction (Wu et al., 2023).

Based on the solar zenith angle information ($ref_{elev}$) in the ICESat-2 ATL03 product, the photon incidence angle ($\theta_1$) can be expressed as follows (Ma et al., 2020):

$$\theta_1 = \frac{\pi}{2} - ref_{elev} \tag{4}$$

Based on Snell's law of refraction, the refraction angle ($\theta_2$) is derived:

$$\theta_2 = \sin^{-1}\left(\frac{n_1 \sin \theta_1}{n_2}\right) \tag{5}$$

where $n_1$=1.00029 and $n_2$=1.34116 denote the refractive indices of air and water, respectively.

Considering the change in their propagation path when photons travel through water, the path length can be expressed as follows:

$$S_1 = \frac{Z_0}{\cos \theta_1}$$
$$S_2 = \frac{S_1 n_1}{n_2} \tag{6}$$

where $S_1$ and $S_2$ represent the underwater path lengths of a photon before and after considering the refractive effect, respectively, and $Z_0$ is the water depth before refraction correction.

Therefore, the difference in photon position owing to refraction ($P$) can be obtained as follows:

$$P = \sqrt{S_2^2 + S_1^2 - 2S_2 S_1 \cos \varphi},$$
$$\varphi = (\theta_1 - \theta_2) \tag{7}$$

Consequently, the difference in the along-track direction ($\Delta x$) and elevational direction ($\Delta d$) due to refraction can be expressed as follows:

$$\begin{cases} \Delta x = P \cos \beta \\ \Delta d = P \sin \beta \\ \beta = \dfrac{\pi}{2} - \theta_1 - \alpha \\ \alpha = \sin^{-1}\left( \dfrac{S_2 \sin \varphi}{P} \right) \end{cases} \quad (8)$$

The datum of water depth is then referenced to the DTU21MSS datum (Wu et al., 2023), as follows:

$$Z_{depth} = h_{\mathrm{MSSH}} - Z_{sentinel} + Z_0 + \Delta d - h_b + \Delta h \quad (9)$$

where $h_{MSSH}$ is the mean sea surface height (MSSH) obtained from the DTU21MSS model; $Z_{sentinel}$ is the SSH at the Sentinel-2 imaging time; $Z_0$ is the ellipsoidal height of a sea-surface photon; $h_b$ is the ellipsoidal height of an underwater photon; and $\Delta h$ is the difference between SSH at the Sentinel-2 imaging time and at the ICESat-2 data acquisition time. The ICESat-2 preprocessing algorithm is illustrated in Figure 5.

Using this protocol, the raw ICESat-2 photons were preprocessed. Two representative samples, over the Dongsha Group and Meijiu Reef, represented by the cyan dotted boxes in Figure 3(b) and (c), respectively, are illustrated (Figure 6). The data acquisition time of the ICESat-2 track for the Dongsha Group was 22:41 pm (nighttime), 29 January, 2019. Photons representing sea-surface and bathymetry information can be clearly distinguished, as both are continuously distributed along the track direction. Notably, only photons with confidence levels of 3 and 4 were used in bathymetric-information extraction (Neumann et al., 2021). The sea surface is smooth over the study area, and the photons are distributed within ±1 m of the sea surface. In contrast, the distribution of the seafloor point-cloud is irregular. Figure 6(a) depicts the raw ICESat-2 photon data, and Figure 6(b) the results of denoising and identification of the sea-surface and seafloor point-clouds. Following noise-threshold estimation and point-cloud removal, the noise in the point-cloud data was effectively suppressed. The bar chart (Figure 6(b), right panel) displays the cumulative along-track density. The point-cloud density was higher at the sea surface than below the surface, and the cumulative along-track point-cloud density exhibits notable peaks. Based on the proposed approach, the sea-surface point data were accurately identified (yellow lines, Figure 6(b)). Moreover, the bathymetry point data (green points, Figure 6(b)) were identified by locating the areas with the highest elevational point density (red circles, Figure 6(b)). Finally, the sea floor was identified via local least-squares fitting. The red scatter points and the blue line in Figure 6(c) represent the refraction-correction results and the fitted seafloor, respectively.

In comparison, the ICESat-2 track for Meijiu Reef (Figure 7) was acquired at 15:26 pm (daytime), March, 9, 2020. Relative to the nighttime results (Figure 6(a)), the daytime results exhibit more noise in the raw ICESat-2 photon data (Figure 7(a)), owing to the greater illumination effects during the day, which presents challenges for water-depth detection. For instance, as shown in Figure 7(b), a significant amount of noise-related point-cloud remained after denoising (e.g., at an along-track distance of ~500 m), although the proposed algorithm effectively identifies the along-track water-depth point-cloud (highlighted by the red circle in Figure 7(b)), and, via a function fitting, achieves robust extraction of the sea floor. The red scatter points and the blue line in Figure 7(c) represent the refraction-correction results and the fitted water depth.

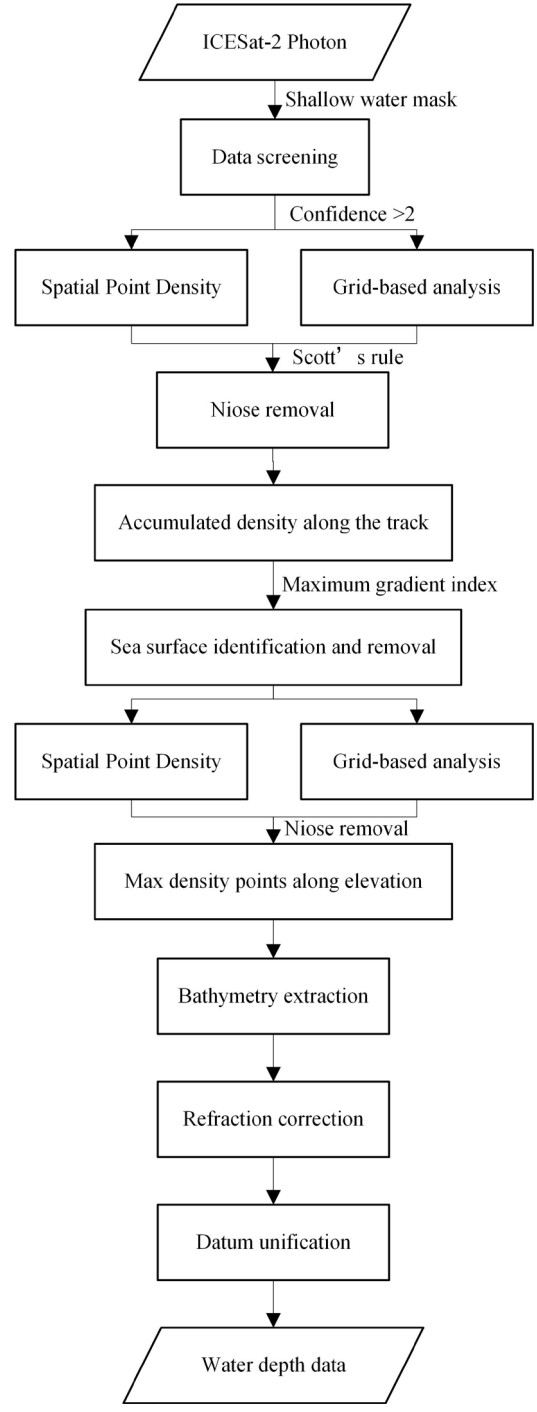

**Figure 5. Flowchart for ICESat-2 water depth extraction.**

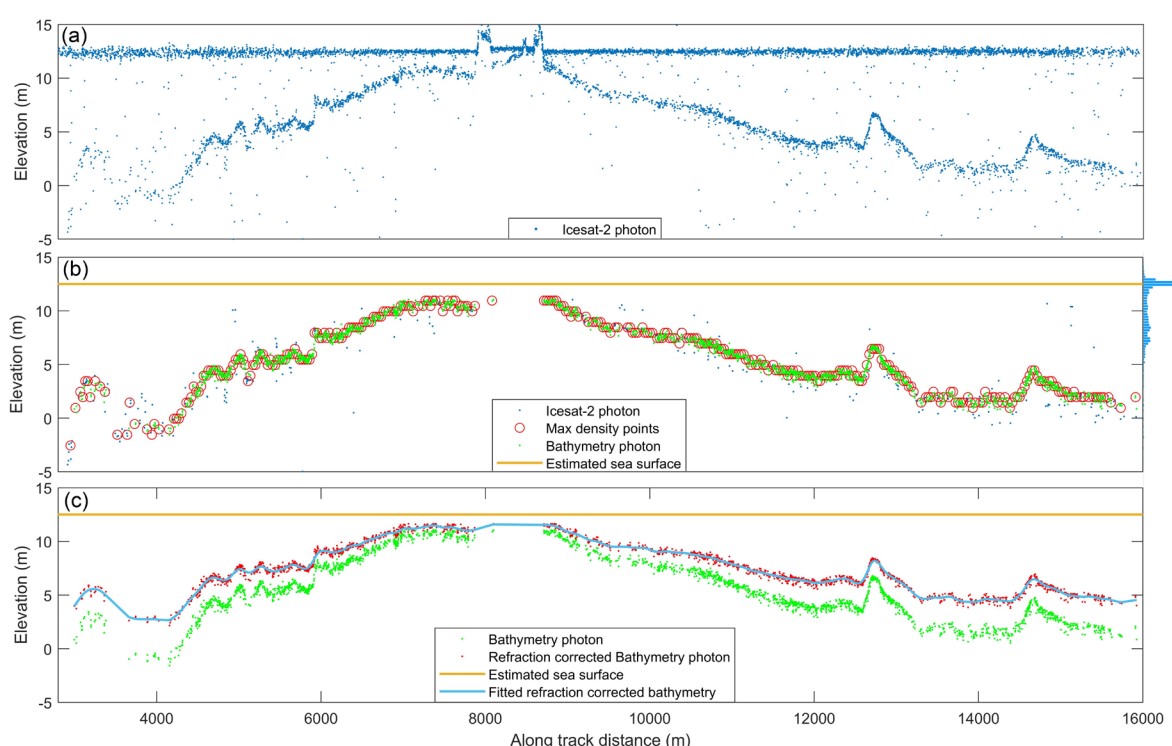

Figure 6. (a) Raw ICESat-2 photons, (b) noise removal, sea surface identification, and water depth extraction, and (c) refraction correction for ICESat-2 track in Dongsha Group (nightime) shown as Figure 3(b).

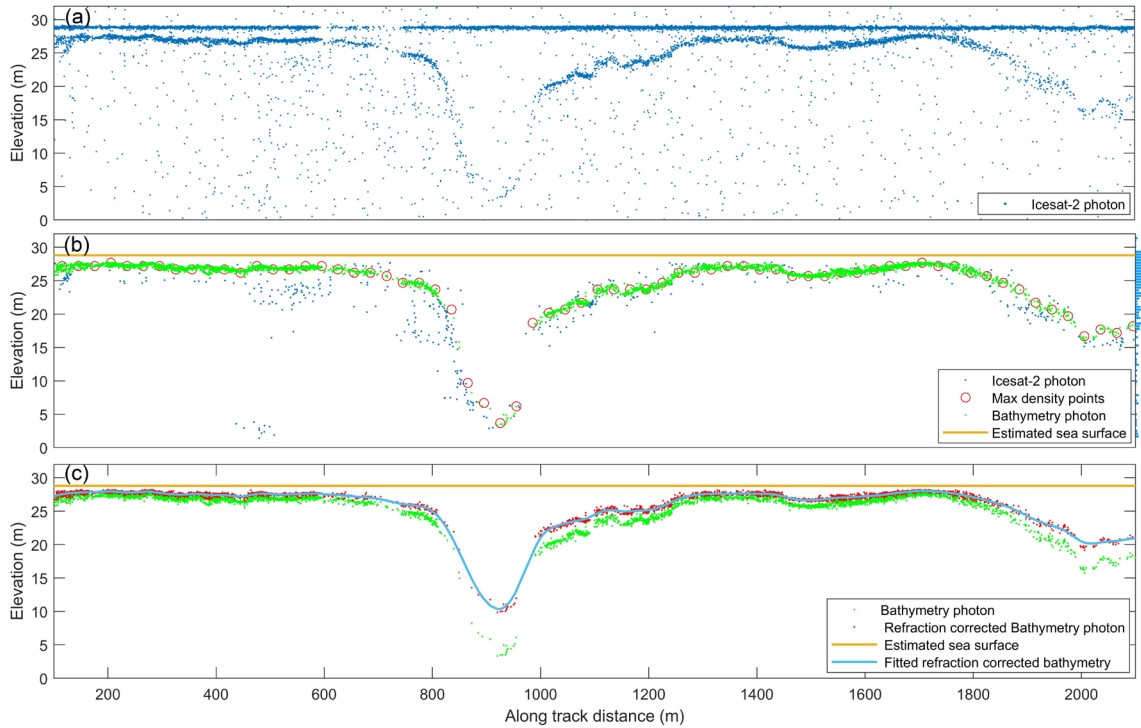

Figure 7. (a) Raw ICESat-2 photons, (b) noise removal, sea surface identification, and water depth extraction, and (c) refraction correction for ICESat-2 track in Meijiu Reef (daytime) shown as Figure 3(c).

## 3.2.    SDB modeling methodology

SDB was modeled by combining the ICESat-2 training data and Sentinel-2 multispectral imagery using the LBM,

which achieves slightly better results than the band-ratio model (Lyzenga et al., 2006; Thomas et al., 2021; Wu et al., 2023). SDB is modeled as follows:

$$H_{\text{SDB}} = h_0 + \sum_{i=1}^{n} h_i \ln[R(\lambda_i) - R_\infty(\lambda_i)] \tag{10}$$

where $H_{\text{SDB}}$ is the water depth derived from a multispectral image; $R(\lambda_i)$ represents the water surface reflectance of band $i$, and $n$ denotes the number of spectral bands used for SDB estimation; and $R_\infty(\lambda_i)$ is the average deep-water reflectance of band $i$. Parameters $h_0$ and $h_i$ are the coefficients estimated via multiple linear regression, as follows:

$$\begin{cases} h_i = \dfrac{\sum\limits_{i=1}^{n} x_i y_i - n\overline{xy}}{\sum\limits_{i=1}^{n} x_i^2 - n\overline{x}^2} \\ h_0 = \overline{y} - h_i \overline{x} \end{cases} \tag{11}$$

where $x_i = R(\lambda_i) - R_\infty(\lambda_i)$, $y_i$ represents the depths obtained from the ICESat-2 training data and $\overline{x}$ and $\overline{y}$ are the mean values of $x_i$ and $y_i$, respectively.

Notably, to ensure robustness and generalizability, the $h_i$ parameters are estimated by all ICESAT-2 water depth training data within the shallow-water area of the reef.

The quality of ICESAT-2 data is affected by water quality, seabed conditions, and inherent noises, which inevitably affects the SDB modeling. Prior to modeling the SDB data, a data-screening scheme based on correlation analysis was applied to eliminate anomalous and noise data. The water depth data for each track was first divided into several segments based on the along-track distance (e.g., 500 m). For each segment, the Pearson correlation coefficient between the ICESat-2 depth and the reflectance of each specific band was calculated, as per (Benesty et al., 2009):

$$\rho(Z, R) = \frac{1}{N-1} \sum_{i=1}^{N} \left( \frac{Z_i - \mu_Z}{\sigma_Z} \right) \left( \frac{R_i - \mu_R}{\sigma_I} \right), \tag{12}$$

where $\rho$ is the Pearson correlation coefficient; $Z$ is the ICESat-2 depth; $R$ is the reflectance; $N$ is the number of ICESat-2 data points in this segment; $\mu_Z$ and $\sigma_Z$ represent the mean and standard deviation of $Z$, respectively; and $\mu_R$ and $\sigma_R$ denote the mean and standard deviation of $R$, respectively.

It is noted that the ICESat-2 data exhibit higher resolution (~0.7 m along-track) than the Sentinel-2 imagery (~10 m). Before correlation analysis, bilinear interpolation was used to estimate reflectance at the locations of ICESat-2 photons. Correlation analysis was conducted track-by-track, and Pearson correlation coefficients were computed for all three visible bands, producing three correlation coefficients for each ICESat-2 photon. An ICESat-2 photon was excluded from SDB training if two or more of its correlation coefficients were smaller than a predetermined threshold (e.g., 0.4).

The deep-water radiative correction was implemented to effectively mitigate sun glint and water column reflectance interferences, resulting in improved performance in reflectance-depth relationship establishment (Jia et al., 2023; Wu et al., 2023). Additionally, the GEBCO_2024 model was used as a reference to select deep-water areas, where regions with depths exceeding 100 m are identified as the deep waters (see the purple rectangles in Fig. 3). For a specific region, the training ICESAT-2 data and corresponding Sentinel-2 imagery were combined to establish a linear relationship between water depth and surface reflectance. Then, multiple linear regression analysis was performed to estimate the $h_i$ coefficients in the LBM.

The Root Mean Square Error (RMSE) and coefficient of determination (R²) were used to evaluate the accuracy of the SDB data and the correlation between the predictions and validation values, as follows:

$$\text{RMSE} = \sqrt{\frac{1}{N}\sum_{i=1}^{N}(y_i - \hat{y}_i)^2},$$

$$R^2 = 1 - \frac{\sum_{i=1}^{N}(y_i - \hat{y}_i)^2}{\sum_{i=1}^{N}(y_i - \overline{y})^2}. \tag{13}$$

where $y_i$ and $\hat{y}_i$ represent the $i$ th estimated depth and the validation data, and $\overline{y}$ denotes the mean value.

While R² is redundant with RMSE, it has additional characteristic of reflecting the correlation between the predictions and validation values. Therefore, RMSE was used as the accuracy metric for the SDB model, and R² values are also included.

## 4. Results and discussion

### 4.1. SDB estimation

SDB modeling was performed using 1298 ICESat-2 shallow-water depth data tracks (2018–2024) and 70 Sentinel-2 images. Functional mapping between the training data and multispectral information was established within the shallow-water mask based on the linear band model approach, where three visible bands (B2, blue; B3, green; and B4, red) were used to train the LBM. The derived SDB model (HHU24SWDSCS) covers 128 islands and reefs in the SCS (Table A1).

Figure 8 illustrates the SDB results of HHU24SWDSCS, showing rich details of seafloor topography. The SDB depth ranges from 0 to 30 m, capturing the typical morphology of coral reefs and sandbanks. In Area 1 (Xisha Islands), it shows water depths ranging from 0 to 15 m. This area contains numerous ring-shaped coral reefs (e.g., the Yongle Group and Huaguang Reef), and the seafloor topography is characterized by deeper central regions and shallower outer regions. In Area 2 (Dongsha Islands), it indicates water depths ranging from 0 to 20 m. Around the outer coral ring reefs, the water depths range from 2 to 10 m, with a gradual deepening towards the west and shallowing towards the east. Within the inner waters, the average depth is ~10 m, with the deepest point reaching 19 m. Areas 3–7 (Nansha Islands) exhibit more diverse depth patterns. The islands and reefs in this region primarily comprise coral reefs and submerged shoals, which are typically small and scattered. The water depths are generally deeper than those in the Xisha Islands and Dongsha Islands, ranging from 0 to 30 m.

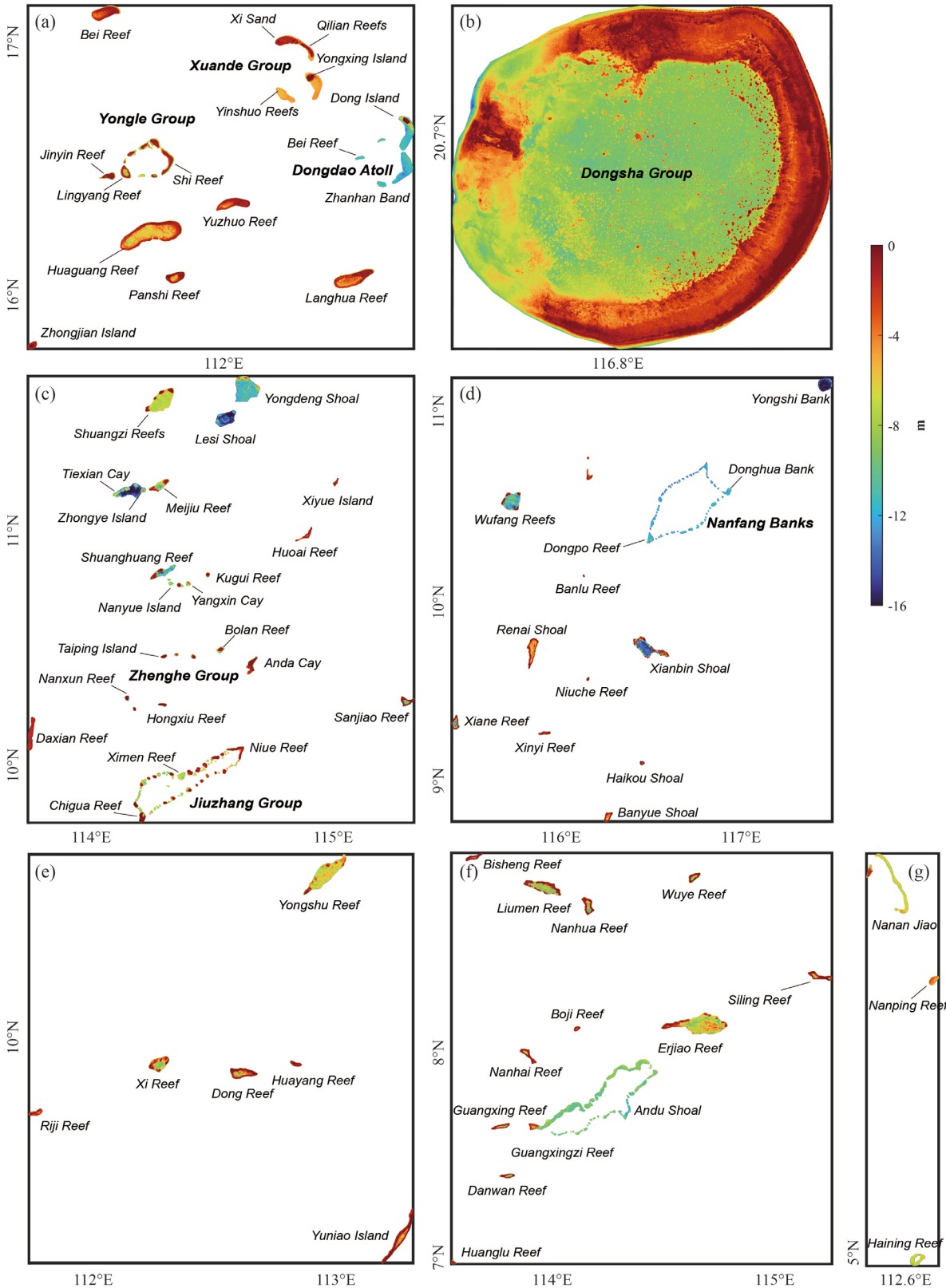

**Figure 8. The SDB results in (a) Area 1, (b) Area 2, (c) Area 3, (d) Area 4, (e) Area 5, (f) Area 6, and (g) Area 7, respectively.**

Figure 9 presents the SDB training results for the entire SCS (Figure 9(a)) and for the seven subareas (Figure 9(b)-(h)). Each subfigure includes annotations for the number of point data, the regression linear equation, as well as R² and RMSE. It is evident that the SDB results are highly consistent with the training data. Regression

analysis of the training data for the entire SCS region yielded an RMSE of 0.89 m. The best performance was achieved for Area 1 (Figure 9 (b)), with an RMSE of 0.64 m. For all of the subareas, the RMSEs are below 1.1 m, i.e., <5% of the maximum detectable water depth in the respective regions. These results reflect the high accuracy of the SDB in fitting this shallow-water bathymetry, with good model stability and robustness.

Figure 10 presents the SDB validation results for the entire SCS (Figure 10(a)) and for the seven subareas (Figure 10(b)-(h)). Noteworthy, the ICESat-2 validation data used here were not introduced in the SDB modeling process, making it suitable for independent validation. The validation results (Figure 10) reveal similar findings to the training results (Figure 9). The validation results for the entire SCS yield an RMSE of 0.82 m, and from 0.53 to 1.24 m in the sub-regions. Comparison of Figure 9 and Figure 10 reveals that the training and validation RMSE are highly consistent. Therefore, this SDB algorithm produces reasonable estimates, exhibiting strong generalization capability and the potential for model transfer.

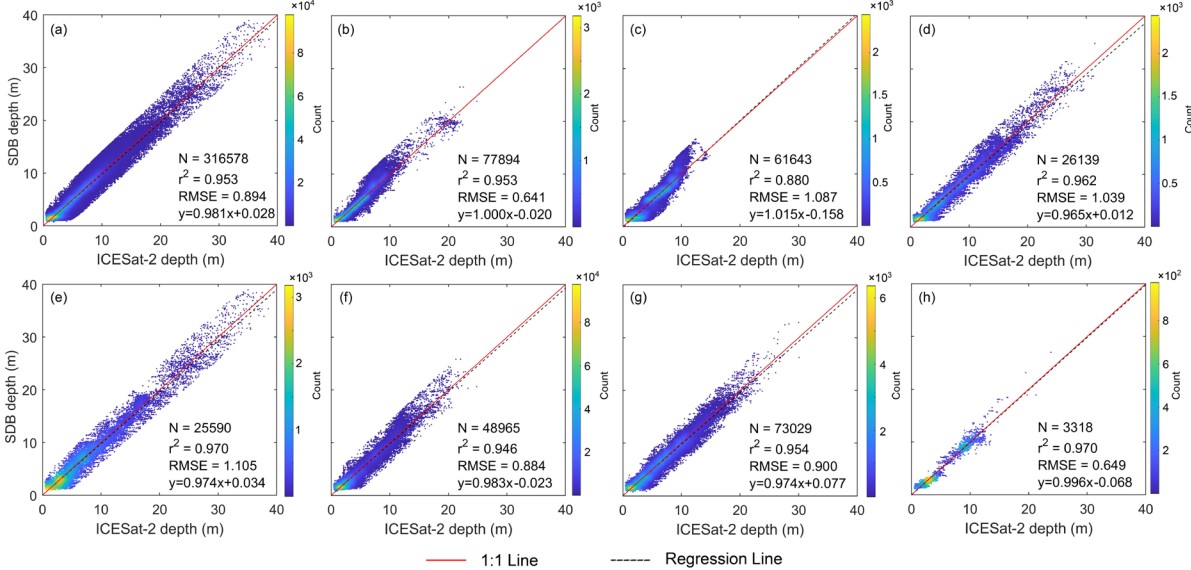

**Figure 9. Training results of HHU24SWDSCS for (a) entire SCS, (b) Area 1, (c) Area 2, (d) Area 3, (e) Area 4, (f) Area 5, (g)Area 6, (h) Area 7, respectively. The red line represents the 1:1 line, and the black dashed line corresponds to the regression line.**

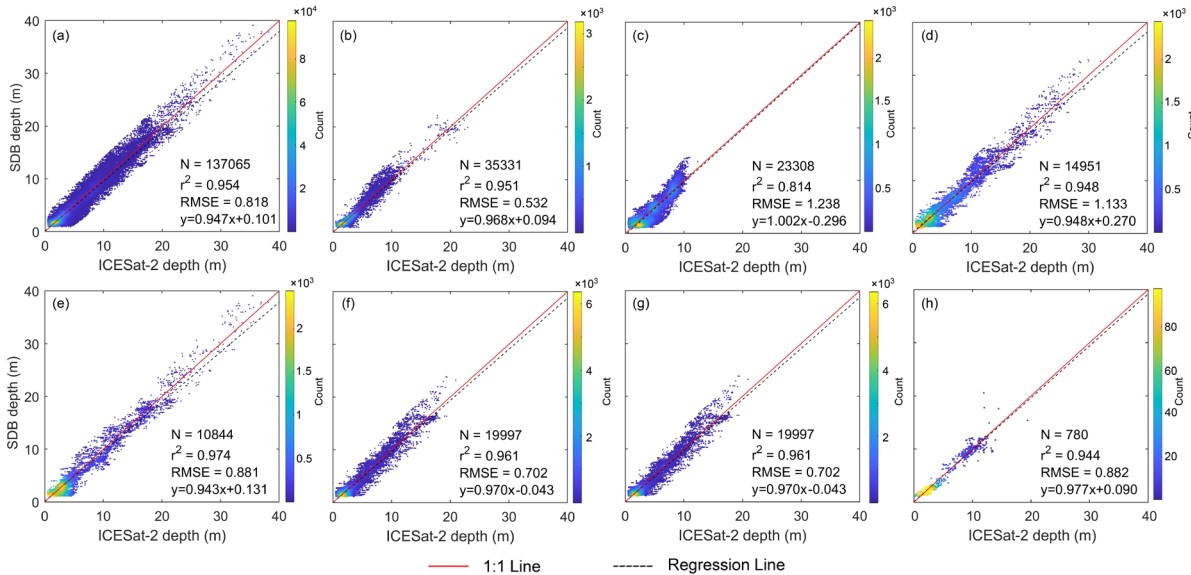

**Figure 10. Validation results of HHU24SWDSCS for (a) entire SCS, (b) Area 1, (c) Area 2, (d) Area 3, (e) Area 4, (f)**

**Area 5, (g)Area 6, (h) Area 7, respectively. The red line represents the 1:1 line, and the black dashed line corresponds to the regression line.**

To illustrate the details of the SDB bathymetry model, six representative reefs shown in Figure 3 were selected for individual analysis. Figure 11 (Column 1) presents the SDB results of these representative islands and reefs, illustrating their geographical distributions, topographical features, and hydrological conditions. The Yongle Group, located in Area 1, comprises several reefs with diameters of ~5 km and average water depths ~5 m (Figure 11(a1)). In addition, Lingyang Reef, situated in the southwestern part of the Yongle Group, exhibits a typical central lagoon morphology, characterized by deeper waters in the center and shallower waters along the edges. The Dongsha Group is located in Area 2, with diameters of over 20 km (Figure 11(b1)). Known for its atoll structure, the Dongsha Islands exhibit a distinct lagoon morphology, and the SDB model accurately captures complex bathymetric patterns, including the central lagoon (~12 m) and the surrounding reef (~3 m). Meijiu Reef, located in Area 3, is a V-shaped reef spanning ~9 km (Figure 11(c1)), with coral reefs primarily in the northeastern and southwestern parts of the island (~3 to 5 m deep) and a central lagoon (up to 20 m deep). Additionally, the SDB model in Figure 11(c1) clearly reveals the complex underwater terrain, including water channels on the western and southwestern sides of the reef. Renai Reef, located in Area 4, is a narrow, elongated north–south reef (Figure 11(d1)); the SDB successfully reveals the narrow passages at the reef's edge and the sharp transitions between the reef flats and the lagoon. Yuniao Reef, located in Area 5, is even narrower and more elongated in a northeast-to-southwest orientation, with its narrowest point being just 1.2 km, presenting a challenge for retrieving effective ICESat-2 water-depth data (Figure 11(e1)); nonetheless, the SDB model still yields reasonable results for this reef, revealing a central lagoon depth of ~8 m and an edge depth of ~4 m. Finally, for Nanhua Reef (located in Area 6), the SDB results (Figure 11(f1)) successfully reveal two water channels approximately 100 m wide in the southwestern and eastern parts of the reef.

Based on the SDB validation results (Figure 11, column 2), most of the discrepancies between the SDB and the ICESat-2 validation data are within ±3 m, with larger discrepancies at the edges of the islands, such as around 20.77°N 116.8°E (Dongsha Group; Figure 11(B2)) and 8.14°N 113.3°E (Yuniao Reef; Figure 11(E2)). These discrepancies can be attributed to two main factors. First, the quality of the ICESat-2 data tends to degrade near boundaries, owing to the complex boundary topography and environmental conditions, thus affecting the accuracy of the depth measurements. Second, there is a significant edge effect in the SDB modeling: as the number of ICESat-2 data-points decreases, the constraints on the linear regression model are reduced and estimation accuracy declines.

The SDB training and validation results are presented in Figure 11, columns 3 and 4, respectively. Based on the training results, for the six representative islands, RMSE from 0.54 to 1.20 m, and the regression slope from 0.90 to 1.05; this reveals high consistency and robust performance using the training dataset. The high $R^2$ values demonstrate a strong correlation between the model predictions and the actual observations, while the low RMSE confirms the accuracy of the model in predicting water depths in shallow areas.

For the six islands, the validation results reveal RMSE of 0.46-1.24 m, and regression slopes of 0.85-1.13. As with the training results, the validation results reveal strong correlation between the model predictions and actual observations. The regression slopes (which are close to unity) indicate that the model performs well not only on training data but also on unseen data, reflecting its robust generalization capability. Along with the results illustrated in Figure 9 and Figure 10, this reveals that the model maintains consistently high performance throughout both the training and validation phases, thus highlighting its stability and reliability.

We next analyzed the SDB validation results for all 128 islands and reefs against the ICESat-2 data. Over 90% of the SDB results exhibit a RMSE consistently below 5% of the maximum depth. The lower modeling accuracy for specific islands and reefs (such as Daxian Reef, Figure 8(c), and Banlu Reef, Figure 8(d)) can be

attributed to the scarcity and uneven distribution of effective water-depth data in the ICESat-2 dataset and to the high level of noise in the images. These results demonstrate that the SDB model effectively captured the fine-scale bathymetric features of shallow-water areas. Incorporating the control data (the ICESat-2 depth data) effectively constrained and enhanced the absolute accuracy of the SDB model. By leveraging the complementary advantages of multi-source remote-sensing data, the precision of the SDB results is ensured.

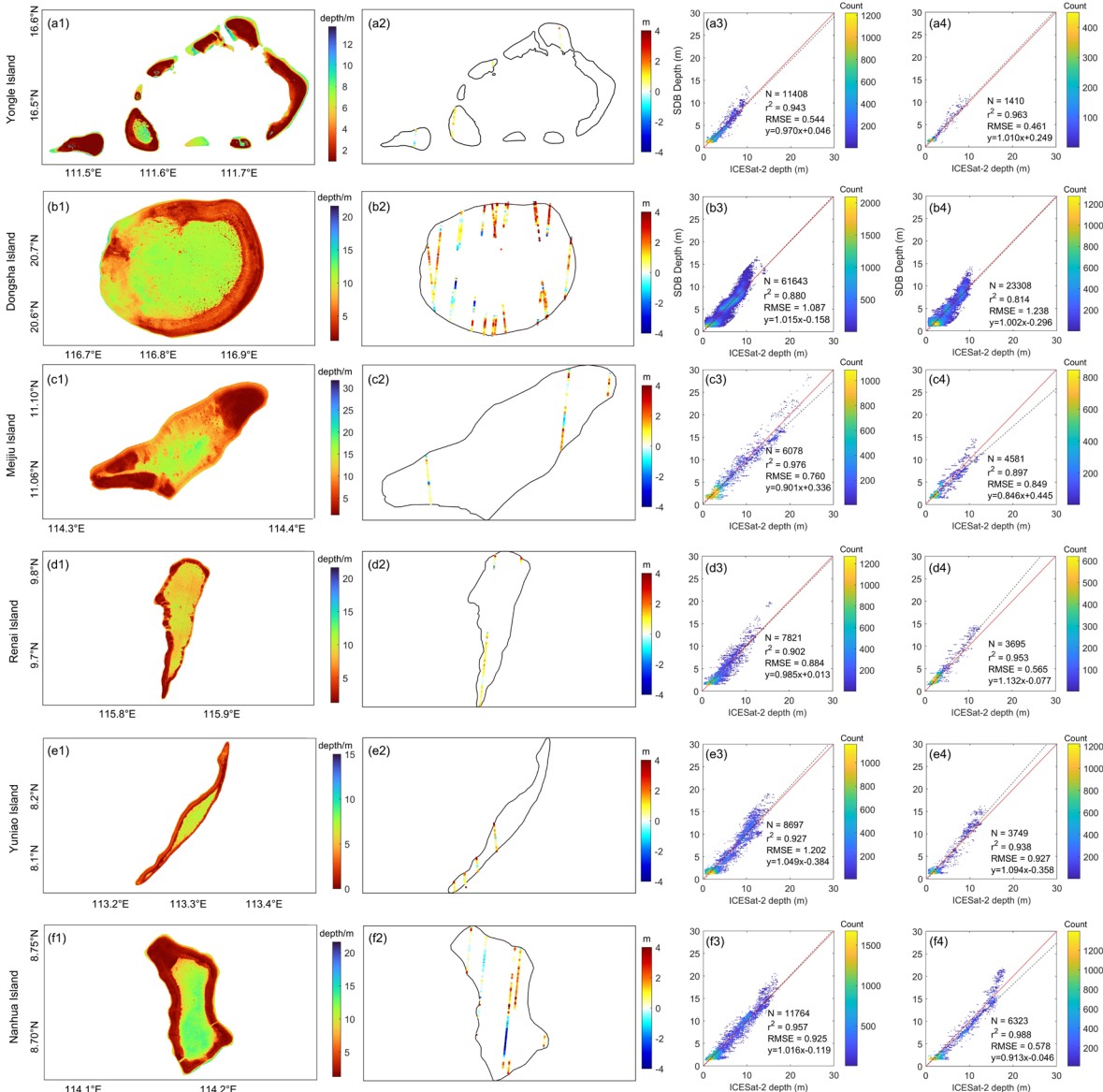

**Figure 11. The HHU24SWDSCS for the representative reefs (first column), the validation results using independent ICESat-2 water depth data (second column), the regression analysis between the SDB results and the training data (third column) and the validation data (fourth column), respectively.**

Using airborne bathymetry data (SIOFM) for the shallow waters near Lingyang Reef, the reef's bathymetry was validated. The latest global bathymetry models, including DTU18BAT (DTU Space), topo_27.1 and SRTM15+ V2.6 (SIO), and GEBCO_2024 (GEBCO Bathymetric Compilation Group), were introduced for validation and analysis. We used nearest-neighbor interpolation to interpolate the bathymetry models to the airborne LiDAR bathymetry points. Based on the validation results (Figure 12), the SDB model achieves notably better estimates than the other models. As shown in Figure 12(a), the differences between the SDB-derived bathymetry and validation data are mostly within ±3 m, whereas the discrepancies exceed 10 m with respect to

the other models. Given that the water depth ranges from 0 to 10 m in the shallow-water areas of Lingyang Reef, this indicates that the existing bathymetry models exhibit relatively poor accuracy and low data reliability in these regions. The RMSE is 1.01 m for the SDB model, as opposed to 61.03 m for DTU18BAT, 25.03 m for topo_27.1, 4.3 m for SRTM15+ V2.6, and 32.38 m for GEBCO_2024 (Table 3). These validation results reveal that the SDB model can provide shallow-water bathymetry with ~1 m accuracy, consistent with the independent ICESat-2-based validation results (Figure 10). More importantly, the SDB model significantly outperforms other existing models for shallow water areas.

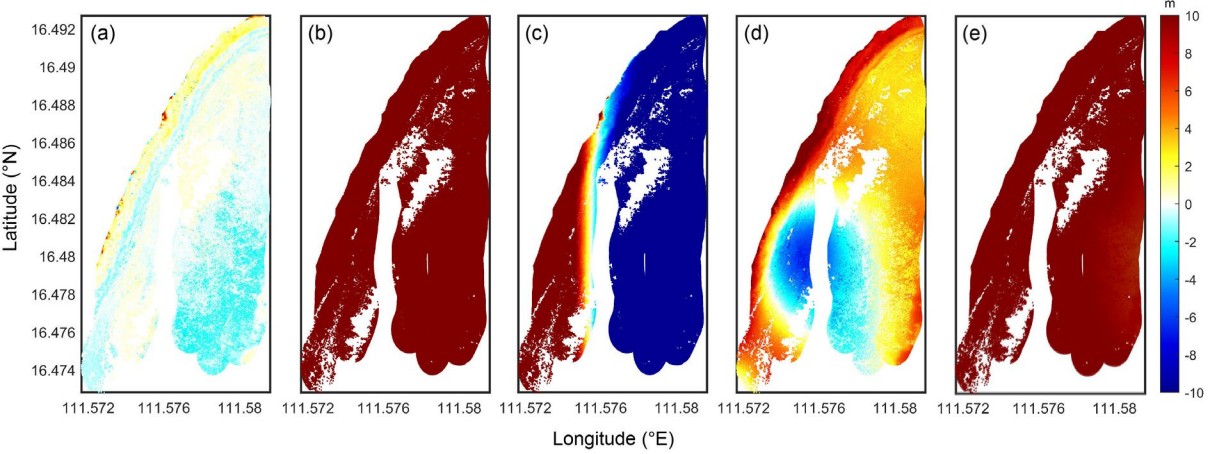

**Figure 12. Validation of (a) HHU24SWDSCS, (b) DTU18BAT, (c) topo_27.1, (d) SRTM15+ V2.6, and (e) GEBCO_2024 against the surveyed airborne LiDAR water depth in Lingyang Reef.**

Table 3. Statistics of the validation results between bathymetry models against airborne LiDAR water depth data (m)

| Models | MAX | MIN | MEAN | RMSE |
|---|---|---|---|---|
| SDB | 3.35 | -4.14 | -0.01 | 1.01 |
| DTU18BAT | 105.76 | 28.68 | 59.25 | 61.03 |
| topo_27.1 | 66.67 | -35.62 | -14.01 | 25.03 |
| SRTM15+ V2.6 | 23.93 | -7.52 | 2.38 | 4.30 |
| GEBCO_2024 | 58.68 | 25.33 | 32.16 | 32.38 |

## 4.2. Discussion

Given that most marine-related production and economic activities are concentrated in shallow-water areas, accurate shallow-water bathymetry has become essential in such activities. Therefore, it is necessary to further evaluate the accuracy of the existing bathymetry models for coastal shallow-water areas. Validation results for the existing bathymetry models for representative reef areas using ICESat-2 data are shown in Figure 13; each column represents the validation results for one model (SDB, DTU18BAT, topo_27.1, SRTM15+ V2.6, and GEBCO_2024 models), and each row, its performance for a specific reef. For all six reefs, the SDB-derived bathymetry results differ from the validation data by <5 m, whereas for the other models, these differences exceed 20 m. Specifically, as shown in Table 4, the SDB validation RMSEs are 0.77 m, 1.25 m, 1.02 m, 1.16 m, 1.35 m and 1.54 m, respectively, whereas for DTU18BAT, topo_27.1, SRTM15+ V2.6, and GEBCO_2024, the RMSEs exceed 2 m, even reaching tens of meters (Table 4). Given that water depth in coastal shallow-water areas is generally <30 m, most of the existing bathymetry models exhibit large uncertainties and low data usability. In contrast, the SDB model achieves relatively robust meter-level accuracy in these regions, demonstrating its superiority in shallow-water bathymetry retrieval. Notably, the existing bathymetry models and

the validation data differ significantly for the Renai Reef area, with a maximum difference exceeding 700 m, while the difference between the SDB results and the validation data for this reef is reduced to the meter level. This discrepancy may because the existing models rely primarily on altimetry-derived gravity anomalies for water-depth data, owing to the scarcity of in situ measurements. However, the poor quality of altimetry data near the coast leads to significant errors in the bathymetry models. Based on the validation results presented in Figure 12 and Figure 13, the SDB bathymetry model achieves high accuracy and robustness in coastal shallow-water areas.

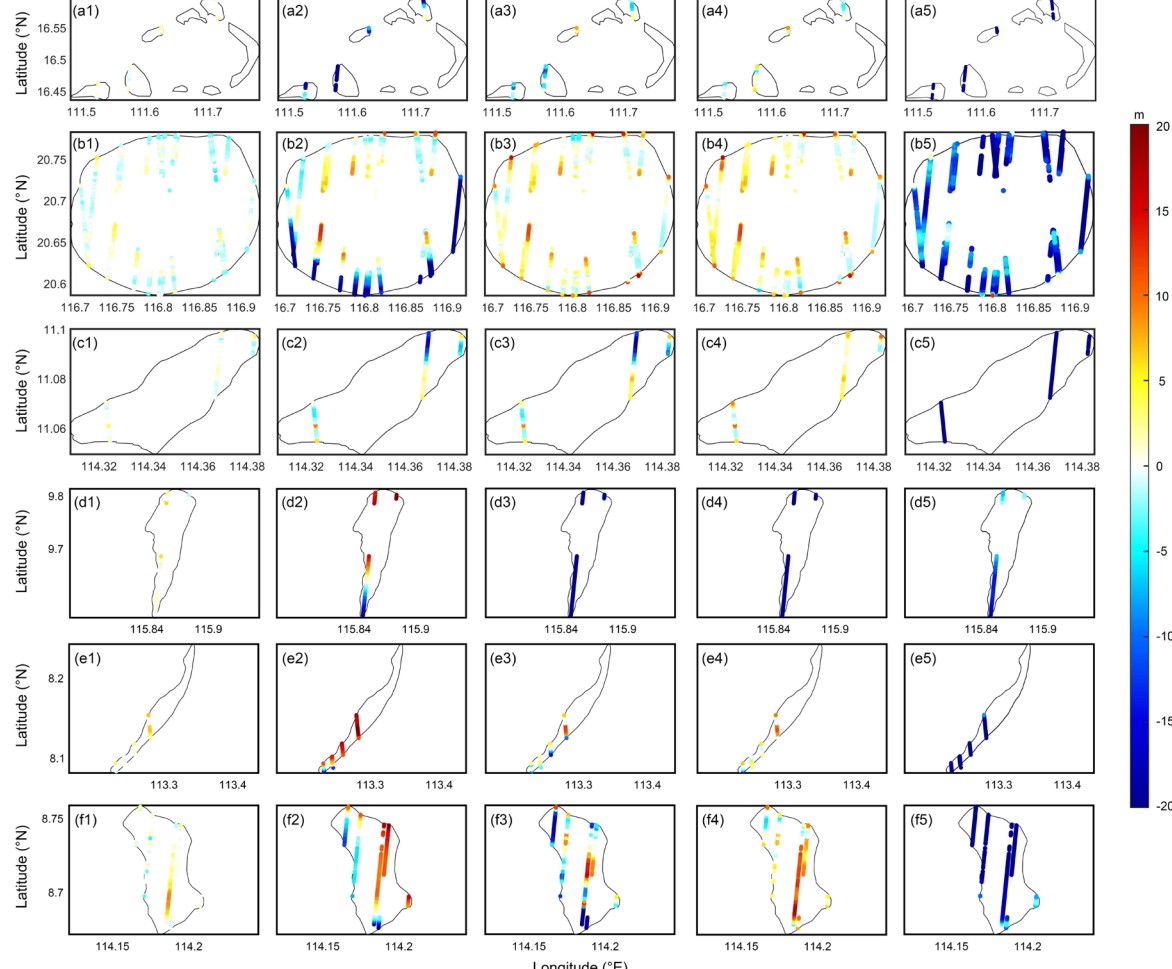

**Figure 13. Validation of the HHU24SWDSCS (first column, a1-f1), DTU18BAT (second column, a2-f2), topo_27.1 (third column, a3-f3), SRTM15+ V2.6 (fourth column, a4-f4), and GEBCO_2024 (fifth column, a5-f5) against independent ICESat-2 water depth data, respectively. The six rows represent the results in the Yongle Island, Dongsha Island, Meijiu Island, Renai Island, Yuniao Island, and Nanhua Island, respectively.**

**Table 4. Statistics on the misfits between different bathymetry models and ICESat-2 validation data over six representative reefs (m)**

| Research areas | Models | Max | Min | Mean | RMSE |
|---|---|---|---|---|---|
| | SDB | 3.49 | -1.31 | 0.65 | 0.77 |
| | DTU18BAT | 2.26 | -91.81 | -48.09 | 56.93 |
| Yongle Group | topo_27.1 | 7.78 | -12.85 | -5.26 | 7.12 |
| | SRTM15+ V2.6 | 8.69 | -5.39 | -0.62 | 2.43 |
| | GEBCO_2024 | -9.5 | -137.84 | -46.07 | 52.35 |
| Dongsha Group | SDB | 8.26 | -7.95 | -0.50 | 1.25 |

| | | | | | |
|---|---|---|---|---|---|
| | DTU18BAT | 13.00 | -92.41 | -5.64 | 12.66 |
| | topo_27.1 | 16.48 | -5.86 | 0.07 | 2.52 |
| | SRTM15+ V2.6 | 16.54 | -7.10 | 0.80 | 2.68 |
| | GEBCO_2024 | 0.11 | -103.94 | -21.81 | 28.11 |
| | SDB | 5.05 | -3.46 | -0.12 | 1.02 |
| | DTU18BAT | 9.39 | -18.80 | -4.28 | 7.43 |
| Meijiu Reef | topo_27.1 | 9.25 | -21.54 | -4.65 | 8.64 |
| | SRTM15+ V2.6 | 10.28 | -2.82 | 1.18 | 2.89 |
| | GEBCO_2024 | -56.65 | -432.38 | -223.53 | 245.51 |
| | SDB | 6.52 | -3.23 | 0.38 | 1.16 |
| | DTU18BAT | -271.25 | -1.129.47 | -682.67 | 727.91 |
| Renai Reef | topo_27.1 | -8.38 | -901.05 | -548.55 | 599.55 |
| | SRTM15+ V2.6 | -42.11 | -888.34 | -526.29 | 573.64 |
| | GEBCO_2024 | -2.11 | -203.03 | -60.19 | 89.47 |
| | SDB | 6.98 | -2.68 | 0.39 | 1.35 |
| | DTU18BAT | 9.85 | -103.13 | -22.44 | 36.34 |
| Yuniao Reef | topo_27.1 | 12.14 | -20.22 | -2.38 | 6.88 |
| | SRTM15+ V2.6 | 11.42 | -14.52 | -1.09 | 5.25 |
| | GEBCO_2024 | -136.75 | -231.43 | -175.62 | 177.45 |
| | SDB | 9.47 | -4.1 | 0.33 | 1.54 |
| | DTU18BAT | -45.36 | -212.57 | -116.03 | 123.41 |
| Nanhua Reef | topo_27.1 | 20.89 | -141.41 | -12.95 | 32.25 |
| | SRTM15+ V2.6 | 17.26 | -6.02 | 1.86 | 5.57 |
| | GEBCO_2024 | -1.77 | -252.42 | -118.55 | 136.19 |

Furthermore, benefitting from the rich spatial information of the SDB model, the spatial detail and accuracy of existing bathymetry models are analyzed. The differences between the SDB results and those of the DTU18BAT, topo_27.1, SRTM15+ V2.6, and GEBCO_2024 models were calculated for the representative reefs (Figure 14). This revealed relatively large differences, with maximum discrepancies exceeding 50 m. Statistically, the RMSE of the differences between the DTU18BAT, topo_27.1, SRTM15+ V2.6, and GEBCO_2024 models against the SDB results reach tens of meters across the six typical island regions, particularly for the Renai Reef area, for which the RMSE exceeds 100 m.

These results reveal that the existing bathymetry models are significantly deficient in spatial resolution, modeling accuracy, and detailed signal depiction for coastal shallow-water areas, making it difficult to meet the current demands of navigation, nearshore economic activities, port construction, and other production activities. The SDB model, which achieves 10 m spatial resolution, meter-level modeling accuracy, detailed bathymetry signals, as well as being efficient and low-cost, therefore constitutes an improvement for coastal shallow-water areas and provides fundamental data support for research in oceanography, geodesy, and other disciplines.

Nonetheless, the sources of error in the SDB results cannot be ignored. First, its accuracy is substantially influenced by water conditions, including turbidity, water type, and water color, which directly affect the underwater light penetration and reflectance measurements of remote-sensing images (Caballero and Stumpf, 2020, 2023). Additionally, although the Sentinel-2 images used in this study have undergone correction for atmospheric effects, residual errors from atmospheric effects, image noise, and the influence of sun glint may still reduce the quality of the SDB results (Warren et al., 2019). The quality of the ICESat-2 data is another key factor in SDB modeling, as its signal-to-noise ratio and limited deep-water penetration capabilities may lead to

insufficient underwater topographic-information retrieval. Regarding the selection of the SDB-estimation methods, the empirical methods rely on a priori depth data to establish the relationship between reflectance and water depth. However, this relationship becomes non-linear in deeper waters, making SDB estimates unreliable in deep-water areas (Ashphaq et al., 2021; Wu et al., 2024b). Furthermore, the impact of changes over time on underwater topography should not be overlooked, as sedimentation, erosion, ocean currents, and human activities can all alter shallow-water bathymetry over time (Caballero and Stumpf, 2021; Niroumand-Jadidi et al., 2020).

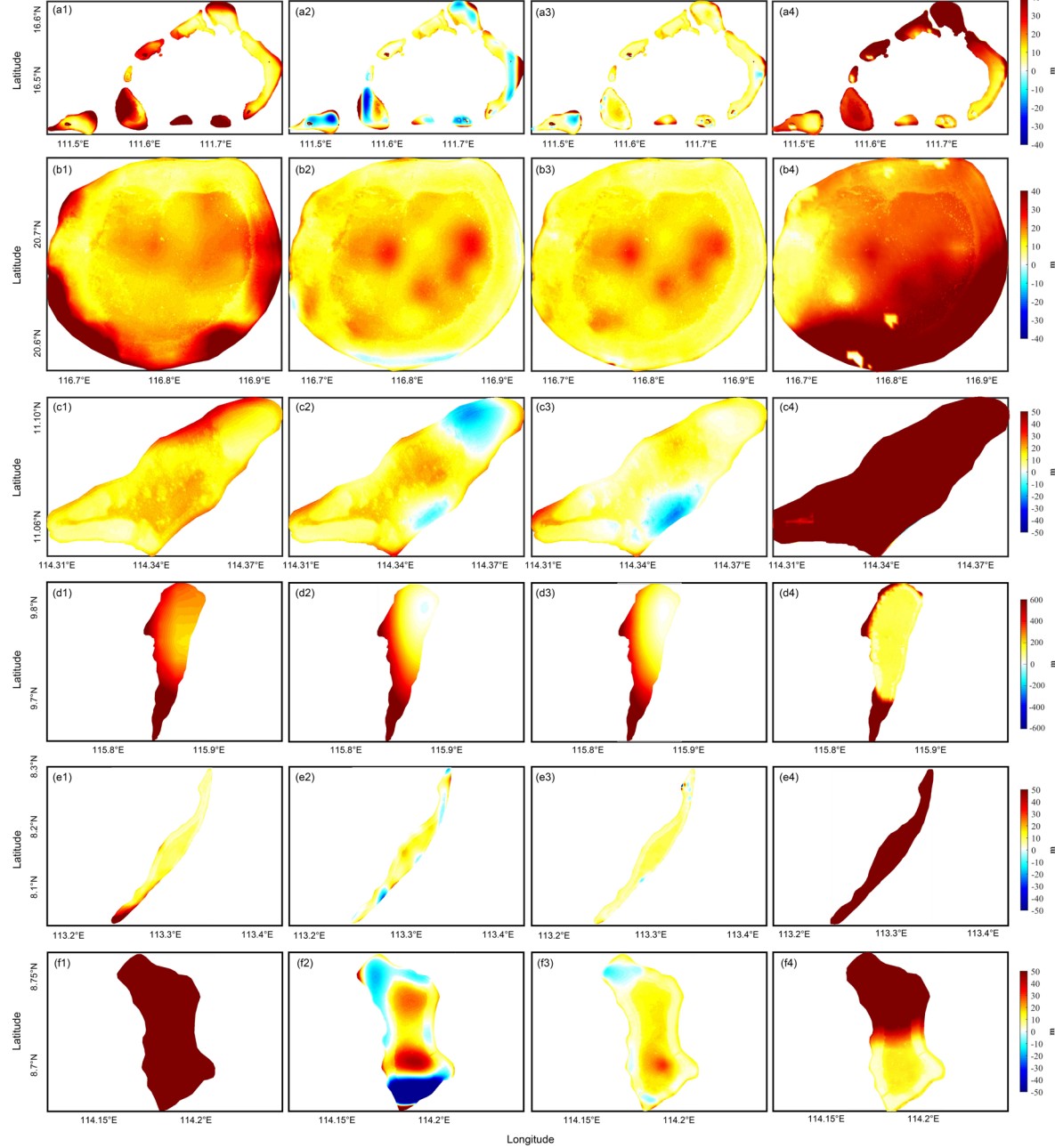

**Figure 14. The comparison of the representative reefs between the SDB results and DTU18BAT (first column), topo_27.1 (second column), SRTM15+ V2.6 (third column), and GEBOCO_2023 (fourth column), respectively.**

**Table 5. Statistics on the misfits between different bathymetry models and SDB results over six representative reefs (m)**

| Research areas | Models | Max | Min | Mean | RMSE |
|---|---|---|---|---|---|
| Yongle Group | DTU18BAT | 137.99 | 1.08 | 22.42 | 30.26 |
| | topo_27.1 | 111.27 | -35.58 | 2.14 | 14.08 |
| | SRTM15+ V2.6 | 89.47 | -16.86 | 5.73 | 8.52 |
| | GEBCO_2024 | 179.45 | -7.33 | 40.74 | 50.91 |
| Dongsha Group | DTU18BAT | 126.39 | 1.06 | 17.64 | 22.34 |
| | topo_27.1 | 27.67 | -7.89 | 10.60 | 11.75 |
| | SRTM15+ V2.6 | 32.65 | -4.03 | 10.35 | 11.26 |
| | GEBCO_2024 | 103.97 | -2.58 | 31.86 | 36.14 |
| Meijiu Reef | DTU18BAT | 43.15 | 2.85 | 14.91 | 16.12 |
| | topo_27.1 | 39.08 | -23.78 | 8.28 | 12.58 |
| | SRTM15+ V2.6 | 36.16 | -25.36 | 5.91 | 9.48 |
| | GEBCO_2024 | 498.06 | 32.92 | 278.12 | 293.10 |
| Renai Reef | DTU18BAT | 1172.07 | 189.36 | 398.25 | 447.78 |
| | topo_27.1 | 946.65 | -10.01 | 273.63 | 355.63 |
| | SRTM15+ V2.6 | 935.88 | 2.44 | 255.41 | 332.35 |
| | GEBCO_2024 | 376.35 | 1.11 | 27.69 | 58.33 |
| Yuniao Reef | DTU18BAT | 113.06 | 1.57 | 11.09 | 17.66 |
| | topo_27.1 | 35.70 | -29.71 | 5.60 | 8.79 |
| | SRTM15+ V2.6 | 41.13 | -117.54 | 5.95 | 7.35 |
| | GEBCO_2024 | 441.01 | -74.58 | 218.29 | 228.07 |
| Nanhua Reef | DTU18BAT | 306.60 | 59.61 | 130.26 | 135.83 |
| | topo_27.1 | 176.34 | -141.35 | -2.99 | 36.47 |
| | SRTM15+ V2.6 | 31.42 | -13.06 | 8.67 | 11.27 |
| | GEBCO_2024 | 268.85 | 1.00 | 67.96 | 95.54 |

## 5. Data availability

The HHU24SWDSCS model is openly accessible at https://doi.org/10.5281/zenodo.13852568 (Wu et al., 2024a). The dataset file (HHU24SWDSCS.nc) includes geospatial information (latitude and longitude), shallow-water depth, and the distribution of the reefs.

## 6. Conclusions

Accurate shallow-water bathymetric data are essential for maritime safety, resource exploration, ecological conservation, and oceanic economic development. To address these requirements, we developed the HHU24SWDSCS model using ICESat-2 data and Sentinel-2 high-resolution multispectral imagery to construct detailed bathymetric maps over 120 islands and reefs in the SCS region. A comprehensive framework was developed for integrating the ICESat-2 data and Sentinel-2 imagery for shallow-water bathymetry modeling.

The accuracy and consistency of the SDB model were evaluated using independent ICESat-2 bathymetry data, which demonstrated robust performance with an RMSE of 0.82 m, underscoring its reliability across the SCS region. Further validation in the Lingyang Reef using the airborne LiDAR bathymetry data revealed that the SDB model achieved superior accuracy (1.01 m) compared to traditional models. Comprehensive validation of the latest bathymetric models (i.e., DTU18BAT, topo27.1, SRTM15+ V2.6, and GEBCO_2024) was conducted against the ICESat-2, airborne LiDAR, and SDB data for the shallow-water regions of representative reefs. This

assessment highlighted significant uncertainties, low spatial resolution, and a lack of details in these existing models in coastal regions and shallow waters. Overall, the SDB model represents a significant advancement in shallow-water bathymetry, offering enhanced accuracy, spatial resolution, and coverage, making it a viable alternative to existing bathymetry models and a powerful tool for marine applications, including coastal construction, ecological conservation, petroleum exploration, and scientific research.

In future, we aim to leverage ICESat-2 and Sentinel-2 data for feature extraction and labeling, utilizing deep learning techniques to construct detailed bathymetric maps of global shallow-water regions. Critical environmental factors such as water quality, water color, seafloor characteristics, and illumination conditions should be considered during model training to enhance generalization capabilities across diverse marine environments. By integrating multiple data sources, including ICESat-2 water-depth data, SDB data, satellite altimetry data, and multibeam sonar sounding, we aim to develop a high-precision, seamless bathymetry model for both shallow and deep waters. Furthermore, the developed SDB technology holds significant potential for coastal and estuarine applications, particularly in monitoring sediment dynamics and investigating temporal variations in intertidal regions.

## Appendix A

**Table A2 Information of the research areas and distribution of islands**

| Research areas | Island groups | Island name | Latitude (°N) | Longitude (°E) | Research areas | Island groups | sea | Latitude (°N) | Longitude (°E) |
|---|---|---|---|---|---|---|---|---|---|
| Area 1 | Xuande Group | Yongxing Island | 16.83 | 112.33 | Area 3 | Jiuzhang Group | Jinghong Island | 9.88 | 114.32 |
| | | Shi Island | 16.85 | 112.35 | | | Nanmen Reef | 9.90 | 114.40 |
| | | Xi Sand | 16.97 | 112.20 | | | Ximen Reef | 9.90 | 114.47 |
| | | Zhaoshu Island | 16.97 | 112.27 | | | Dongmen Reef | 9.92 | 114.50 |
| | | Bei Island | 16.97 | 112.30 | | | Anle Reef | 9.93 | 114.52 |
| | | Zhong Island | 16.95 | 112.32 | | | Changxian Reef | 9.93 | 114.55 |
| | | Nan Island | 16.93 | 112.33 | | | Zhuquan Reef | 9.95 | 114.57 |
| | | Bei Reef | 16.93 | 112.33 | | | Niue Reef | 9.97 | 114.62 |
| | | Zhong Reef | 16.93 | 112.33 | | | Ranqing Reef | 9.88 | 114.60 |
| | | Nan Reef | 16.92 | 112.33 | | | Ranqing Sand | 9.90 | 114.57 |
| | | Dongxin Reef | 16.92 | 112.35 | | | Longxia Reef | 9.88 | 114.53 |
| | | Xixin Reef | 16.92 | 112.35 | | | Bianshen Reef | 9.87 | 114.52 |
| | Dongdao Atoll | Dong Island | 16.67 | 112.73 | | | Zhangxi Reef | 9.83 | 114.47 |

| Area | Group | Name | Lat | Long | | Area | Group | Name | Lat | Long |
|---|---|---|---|---|---|---|---|---|---|---|
| | | Gaojian Reef | 16.57 | 112.63 | | | | Quyuan Reef | 9.80 | 114.40 |
| | | Beibian Reef | 16.53 | 112.55 | | | | Qiong Reef | 9.75 | 114.35 |
| | | Zhanhan Band | 16.42 | 112.62 | | | | Chigua Reef | 9.70 | 114.28 |
| | | Langhua Reef | 16.05 | 112.55 | | | | Guihan Reef | 9.77 | 114.25 |
| | | Ganquan Island | 16.50 | 111.58 | | | | Hua Reef | 9.85 | 114.27 |
| | | Shanhu Island | 16.53 | 111.60 | | | | Jiyang Reef | 9.87 | 114.28 |
| | | Jinyin Island | 16.43 | 111.50 | | | | Huoai Reef | 10.88 | 114.93 |
| | | Zhenhang Island | 16.45 | 111.70 | | | | Xiyue Island | 11.07 | 115.02 |
| | | Guangjin Island | 16.45 | 111.70 | | | | Daxian Reef | 10.07 | 113.87 |
| | | Jinqing Island | 16.45 | 111.73 | | | | Sanjiao Reef | 10.17 | 115.32 |
| | | Lingyang Reef | 16.45 | 111.58 | | | | Antang Reef | 10.88 | 116.43 |
| | Yongle Group | Quanfu Island | 16.57 | 111.67 | | | | Donghua Reef | 10.55 | 116.93 |
| | | Yagong Island | 16.57 | 111.68 | | | | Wufang Reef | 10.48 | 115.75 |
| | | Yin Reef | 16.58 | 111.70 | | | | Wufangnan Reef | 10.45 | 115.77 |
| | | Yinyuzi Island | 16.58 | 111.70 | | | Wufang Group | Wufangwei Reef | 10.47 | 115.72 |
| | | Xianshe Reef | 16.55 | 111.72 | | | | Wufangxi Reef | 10.50 | 115.70 |
| | | Kuangzi Sand | 16.45 | 111.63 | | Area 4 | | Wufangbei Reef | 10.53 | 115.72 |
| | | Shi Reef | 16.55 | 111.75 | | | | Wufangtou Reef | 10.53 | 115.78 |
| | | Huaguang Reef | 16.20 | 111.67 | | | | Banlu Reef | 10.13 | 116.13 |
| | | Yuzhuo Reef | 16.33 | 112.02 | | | | Yongshi Bank | 11.08 | 117.47 |
| | | Panshi Reef | 16.05 | 111.77 | | | | Xiane Reef | 9.35 | 115.43 |
| | | Bei Reef | 17.08 | 111.50 | | | | Xinyi Reef | 9.33 | 115.95 |
| | | Zhongjian Reef | 15.78 | 111.20 | | | | Haikou Shoal | 9.18 | 116.45 |
| Area 2 | Dongsha Group | Dongsha Island | 20.72 | 116.70 | | | | Banyue Shoal | 8.87 | 116.27 |
| | | Dongsha | 20.67 | 116.90 | | | | Yenai Reef | 9.72 | 115.88 |

| Area | Group | Reef | Lat | Long |
|---|---|---|---|---|
| | | Reef | | |
| | Shuangzi Reefs | Gongshi Reef | 11.47 | 114.40 |
| | | Beizi Island | 11.45 | 114.35 |
| | | Beiwai Reef | 11.45 | 114.35 |
| | | Nanzi Island | 11.43 | 114.33 |
| | | Nailuo Reef | 11.38 | 114.30 |
| | | Dongnan Shoal | 11.40 | 114.37 |
| | | Dongbei Shoal | 11.43 | 114.40 |
| | | Beizi Shoal | 11.43 | 114.38 |
| Area 3 | Zhongye Group | Zhongye Island | 11.05 | 114.28 |
| | | Tiezhi Reef | 11.08 | 114.38 |
| | | Meijiu Reef | 11.05 | 114.32 |
| | | Tiexian Reef | 11.07 | 114.23 |
| | Daoming Group | Shuanghuang Reef | 10.70 | 114.32 |
| | | Nanyue Island | 10.67 | 114.42 |
| | | Yangxin Cay | 10.70 | 114.52 |
| | | Kugui Reef | 10.77 | 114.58 |
| | Zhenghe Group | Taiping Island | 10.38 | 114.37 |
| | | Zhong Bank | 10.37 | 114.38 |
| | | Zhongzhou Reef | 10.38 | 114.42 |
| | | Chunqian Sand | 10.38 | 114.47 |
| | | Bolan Reef | 10.42 | 114.58 |
| | | Anda Reef | 10.35 | 114.70 |
| | | Hongxiu Island | 10.18 | 114.37 |
| | | Nanxun | 10.20 | 114.23 |

| Area | Group | Reef | Lat | Long |
|---|---|---|---|---|
| | | Xianbin Reef | 9.73 | 116.57 |
| | | Niuche Reef | 9.60 | 116.17 |
| Area 5 | | Yongshu Reef | 9.58 | 112.97 |
| | Yinqing Group | Xi Reef | 8.87 | 112.23 |
| | | Dong Reef | 8.83 | 112.58 |
| | | Huayang Reef | 8.88 | 112.85 |
| | | Yuniao Reef | 8.27 | 113.37 |
| | | Riji Reef | 8.67 | 111.67 |
| | | Wunie Reef | 8.87 | 114.65 |
| | | Nanhua Reef | 8.75 | 114.18 |
| | | Liumen Reef | 8.83 | 113.98 |
| | | Bisheng Reef | 8.97 | 113.67 |
| | | Erjiao Reef | 8.20 | 114.70 |
| | Yuya Group | Langkou Reef | 8.13 | 114.55 |
| Area 6 | | Xiantou Reef | 8.13 | 114.80 |
| | | Guangxingzi Reef | 7.62 | 113.93 |
| | | Siling Reef | 8.37 | 115.23 |
| | | Boji Reef | 8.10 | 114.13 |
| | | Guangxing Reef | 7.63 | 113.80 |
| | | Nanhai Reef | 7.98 | 113.88 |
| | | Danwan Reef | 7.37 | 113.83 |
| | | Huanglu Reef | 6.95 | 113.58 |
| | | Nantong Reef | 6.33 | 113.23 |
| Area 7 | | Nanan Reef | 5.53 | 112.58 |

| | | | | | | |
|---|---|---|---|---|---|---|
| Reef | | | | | | |
| Yongdeng Shoal | 11.40 | 114.67 | | Nanping Reef | 5.37 | 112.63 |
| Lesi Shoal | 11.33 | 114.62 | | Haining Reef | 4.95 | 112.62 |

# Author contributions.

YW and HS presented the algorithm and carried out the experimental results. YW, HS and DJ prepared the paper and figures with contributions from all the co-authors. YW, HS, DJ, OA, XH and ZL polished the entire manuscript. YL, SC, XS, YS, SD and YC downloaded ICESat-2 and Sentinel-2 data and other products in this work. All authors checked and gave related comments for this work.

# Competing interests.

The contact author has declared that none of the authors has any competing interests.

# Acknowledgements

The Mapper5000 airborne LiDAR bathymetric dataset are provided by the Shanghai Institute of Optics and Fine Mechanics, Chinese Academy of Sciences. DTU18BAT provided by DTU Space is available on https://ftp.space.dtu.dk/pub. The topo_27.1 and SRTM15+ V2.6 provided by Scripps Institution of Oceanography are freely available on https://topex.ucsd.edu/pub/global_topo_1min/. The GEBCO_2024 provided by GEBCO GROUP is available on https://www.gebco.net/data_and_products/gridded_ bathymetry_data/. The authors would also like to thank NASA's National Snow and Ice Data Center for providing the ICESat-2 data (https://nsidc.org/data/icesat-2/data) and the Sentinel Scientific Data Hub of the European Space Agency for providing the Sentinel-2 data (https://browser.dataspace.copernicus.eu/).

# Financial support

This work was supported by the National Natural Science Foundation of China (No. 42374099), the Jiangsu Funding Program for Excellent Postdoctoral Talent under Grant 2024ZB632, the Fundamental Research Funds for the Central Universities (B240201097), the State Scholarship Fund from Chinese Scholarship Council (No. 201306270014, 202006710169), the project of China Railway Corporation (No. 2021-key-14、2021-major-08), and the joint planning of technology and water conservancy of Jiangxi Province (2022KSG01009).

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
