# Peer review of "HHU24SWDSCS: A shallow-water depth model over island areas in South China Sea retrieved from Satellite-derived bathymetry"

_Earth System Science Data, 2024_

## Author Comment (AC1)

The paper uses ICESat data for a training data set for a linear model (multiple linear regression) for SDB from Sentinel-2. The paper shows good results for estimating SDB for several areas in the South China sea. Demonstrations of methods for applying ICESat data for tuning, and for execution are a useful addition to the SDB field.

The methods section has insufficient information to reproduce the application to the satellite. That is necessary, given this is a methods paper. The most noteworthy is how the apparently multiple h_0 and h_i coefficients developed for each ICESat trackline were applied around an island. This should come right after line 340. This is non-trivial, there are 3 h_i coefficients (3 bands).

Response: The authors thank the reviewer for these beneficial comments. The reviewer raised a critical point regarding the determination of regression coefficients in the proposed algorithm, which directly impacts the accuracy of the satellite-derived bathymetry (SDB) modeling. Indeed, the explanation of how the $h_i$ parameters are derived was not sufficiently clear in the original manuscript. In short, to ensure robustness and generalizability, the $h_i$ parameters were derived by constructing a multivariate linear regression (linear band model, LBM) using all available ICESat-2 water depth data within the shallow-water area of a specific reef.

Sentinel-2 multispectral imagery is subject to influences such as illumination conditions, cloud cover, and image noise. Additionally, variations in water quality and bottom composition across different islands necessitate the use of external in situ data for calibration. Consequently, the SDB modeling in this study was conducted for individual islands or closely adjacent archipelagos, under the assumption that illumination and other environmental conditions are consistent within these localized areas. To model the SDB in the island region, the ICESat-2 data was initially acquired by applying a shallow water mask to all available data. Subsequently, the data underwent a series of processing steps, including water depth identification, denoising, refraction correction, and reference datum unification, to obtain high-precision shallow water depth point cloud data. Next, the optimal Sentinel-2 imagery during the ICESat-2 data acquisition period was selected using the AI Earth platform, from which the red, green, and blue band (RGB) reflectance information within the shallow water mask was extracted and corrected based on reference water depth. The reference depth correction is essential to eliminate the radiative energy contributions from the water surface (e.g., sun glint effects), please refer to Q4 for detailed explanation.

Subsequently, the RGB reflectance data was matched with the water depth point cloud. Given that the spatial resolution of ICESat-2 (~0.7 m) is significantly higher than that of the Sentinel-2 imagery (~10 m), the authors calculated the mean depth value of point cloud data located within the same pixel of the Sentinel-2 imagery as prior depth data. In this way, the prior water depth and its corresponding RGB reflectance information were obtained to construct an accurate mapping function. Next, a multiple linear regression analysis was performed based on the prior water depth and the reflectance data, yielding the parameters $h_i$ ( $h_0, h_1, h_2$ ). To enable model evaluation, only a subset of the ICESat-2 depth data was selected for training (~80% of all available ICESat-2 data), while the remaining data was used for validation, ensuring good spatial distribution of the land. Finally, the trained $h_i$ coefficients were applied to the entire shallow water region's RGB reflectance data, resulting in a SDB model for the shallow water area around the island reef.

To provide a more intuitive presentation of the SDB data preparation and modeling process, the Yongle island (Fig. 3(a) in the original text) is selected as a case study. As shown in Fig. 1 below, the Sentinel-2 imagery (background), ICESat-2 water depth (tracks), shallow water mask (white polygons) of the Yongle island region is displayed. Fig. 2(a) presents the specific ICESat-2 water depth distribution, while Fig. 2(b)-(d) shows the corresponding RGB reflectance data, respectively, all of which have undergone Log10 transformation. Using the LBM multiple linear regression model, the authors integrated the ICESat water depth and RGB reflectance data for regression training, yielding the estimated parameters $h_i$: 21.0606, 8.4586, -10.3347, and -0.4124. Finally, these trained $h_i$ parameters were applied to the Sentinel-2 reflectance data within the mask region, resulting in the SDB model for the Yongle island.

Based on the reviewer's comments, the authors have added more detailed information on the modeling approach, the LBM model, and the solution for the $h_i$ parameters in the revised manuscript. Please refer to lines 245-247, line 362, and line 384-386.

[Figure]

Fig. 1 the Sentinel-2 imagery (background), ICESat-2 water depth (tracks), shallow water mask (white polygons) of the Yongle island region

[Figure]

Fig. 2 (a) ICESat-2 bathymetry data and (b)-(d) its corresponding Sentinel-2 reflectance information (RGB)

Also, the limitations of some of the statistical validation need to be identified.

Response: The authors thank the reviewer for these beneficial comments. The authors agree with the reviewer that $R^2$ cannot be used as a rigorous index to evaluate the SDB model, but only for correlation analysis. Based on the reviewer's comments, the authors use RMSE instead of $R^2$ for performance analysis in the whole manuscript. For more explanation, please refer to Q10 as well as the revised manuscript line 392-394 for more information.

Methods:

Q1. The Sentinel-2 reflectance source and atmospheric correction are not explained.

Response: The authors thank the reviewer for these beneficial comments. In the original manuscript, the authors did not provide a clear introduction of the source of Sentinel-2 data and the corresponding corrections applied. The authors used Sentinel-2 L2A products provided by the European Space Agency (ESA) through the AI EARTH platform (engine-aiearth.aliyun.com) as the hyperspectral imagery data source. This imagery includes spectral information from 13 bands, including the red, green, and blue bands.

The ESA provides two levels of Sentinel-2 products: Level-1C (L1C) and Level-2A (L2A). The L1C product is an atmospheric top-of-atmosphere reflectance (TOA) product that has been corrected for radiometric measurements and geometric errors. These corrections include orthorectification and spatial registration using a global reference system (UTM projection combined with the WGS84 ellipsoid), with meter-level accuracy. The L2A product provides images that have undergone orthorectification to derive bottom-of-atmosphere reflectance (BOA). In this study, the authors utilized Sentinel-2 L2A images for SDB modeling, which is based on atmospheric correction of the L1C product to provide surface reflectance data. When selecting imagery, the authors chose high-quality, cloud-free images where the study area was clearly visible and free of obstructions. Additionally, the selected imagery was as close as possible in time to the ICESat-2 data acquisition, ensuring that the underwater topography remained almost unchanged. Although sea floor may change due to factors such as wind, waves, and tides, the variation is minimal and, therefore, the authors have neglected these changes over time in this study.

In addition, the geometric configuration of Sentinel-2 satellite imagery introduces sun glint contamination in deep-water regions, potentially compromising shallow-water energy extraction. To mitigate this effect, we implemented sun glint correction across RGB bands using deep-water reflectance information, with detailed methodological procedures available in (Wu et al., 2023; Jia et al., 2023). Based on the reviewer's comments, the authors have included additional explanations on the Sentinel-2 data in the revised manuscript, please refer to lines 159-166 for detailed information.

Q2. Was one image used for each location?

Response: The authors thank the reviewer for these beneficial comments. Yes. For each island or adjacent reef group, the authors selected the optimal Sentinel-2 image within the ICESat-2 data acquisition period for SDB modeling. Given that the Sentinel-2 image width is ~290 km, some islands are located close enough together that multiple islands or reefs (e.g., Fig. 3 in the original manuscript) may appear within the same image. However, considering potential interference from factors such as sun glint, cloud cover, and speckle noise in Sentinel-2 imagery, as well as variations in water quality and bottom conditions across different regions, the authors chose the best-quality image available for each specific island region within the ICESat-2 data period to ensure accurate SDB modeling.

Furthermore, Fig. 3 and Table 2 in the original manuscript present typical island and reef regions selected by the authors in the Xisha and Nansha areas. Table 2 show the specific image IDs and ICESat-2 track numbers used for SDB modeling in each region. In total, the authors utilized 70 images for the SDB modeling process across more than 120 islands and reefs in the South China Sea. Based on the comments of the reviewer, the authors have provided additional explanations in the revised manuscript to clarify the factors influencing the selection of images for SDB modeling. Please refer to lines 170-178 and lines 239-243 for more information.

Q3. The selection of areas for the regression is not clear. Perhaps because text is split up between lines 170 and 320-335.

Response: The authors thank the reviewer for these beneficial comments. The authors realize that the selection of the modeling regions for SDB regression in the original manuscript was not clearly presented, which may cause confusion for the readers. In this study, the shallow water areas surrounding the islands and reefs were chosen as the SDB modeling regions, as shown in Figs. 1 and 8 in the original manuscript. Due to limitations in image quality and water quality conditions, the authors restricted each SDB modeling region to either a single island or a group of nearby reefs. For example, in Fig. 3(a) and Table 2 of the original manuscript, typical island regions in the Xisha and Nansha areas are presented, along with the images and ICESat-2 data used for SDB modeling in each region. Taking the Yongle Atoll as an example (Fig. 3 below), it includes several small islands such as Ganquan Island and Jinyin Island. Due to the relatively concentrated spatial distribution of these islands, the authors assumed that the Sentinel-2 image quality, lighting conditions, and other factors would be similar across these regions. As such, a unified regression model was applied to the entire group of islands in the region. The authors first selected the high-quality Sentinel-2 L2A image (20210817T025549) and extracted the RGB band information within the shallow water mask region. Additionally, as confirmed in the study by Jia et al. (2023), applying deep-water reflectance correction to Sentinel-2 imagery improves the accuracy of SDB modeling. Based on the GBCO empirical model, the authors identified regions with water depths exceeding 100 m in the modeling area. The RGB reflectance of these deep-water regions was extracted and used as correction information to eliminate sun glint and other surface effects in the RGB data.

[Figure]

Fig. 3 the Sentinel-2 imagery (background), ICESat-2 water depth (tracks), shallow water mask (white polygons) of the Yongle island region

Next, the authors used 19 ICESat-2 tracks containing sea floor information as prior water depth data, which underwent processing steps including water depth point cloud extraction, noise removal, refraction correction, and reference datum unification. Due to the presence of significant Gaussian noise in the depth point cloud, the authors applied piecewise linear fitting to identify the mid-line of the water depth point cloud, which was then used as the true water depth. The depth range was subsequently gridded, and the fitted water depth values were uniformly sampled within grids of different depths, ensuring an equal distribution of point cloud data across varying depth levels. Additionally, as the ICESat-2 data has a significantly higher along-track resolution (0.7 m) compared to the Sentinel-2 spatial resolution (10 m), the authors computed the average depth for the ICESat points located within each pixel and used this averaged value as the prior water depth for that pixel. This process ensures that the prior depth data is consistent with the spatial resolution of the Sentinel-2 imagery.

Finally, the authors extracted the RGB reflectance information from each Sentinel-2 pixel along the ICESat-2 track and performed multiple linear regression fitting using the LBM model, relating this reflectance data to the ICESat-2 prior water depth. Through this process, the regression parameters were estimated. The trained model was then applied to the Sentinel-2 reflectance data, resulting in the SDB model for the shallow water region of the Yongle island.

It is important to note that the islands and reefs mentioned in line 170 of the original manuscript were selected as typical examples from the Xisha and Nansha regions to facilitate the demonstration of the SDB modeling process and its details. The regression modeling areas for these islands are located within the shallow water mask regions shown in Fig. 3 of the original manuscript (inside the white shapes). For all other islands, the corresponding regression modeling areas can be found in the shallow water regions of Fig. 1 of the original manuscript (inside the gray shapes). Additionally, the content in lines 320-325 focuses on selecting the ICESat-2 water depth data and Sentinel-2 RGB reflectance information with the highest correlation based on Pearson correlation analysis. This step is critical to avoid including outliers or noise that could affect the SDB modeling. Based on the reviewer's comments, the authors have further clarified the issue of regression modeling areas in the revised manuscript and have expanded the explanations related to data selection and the modeling approach, making the manuscript clearer, please refer to lines 173-179, and line 189-196 for more information.

Q4. Line 170   What does this mean?   "We used the GEBCO_2023 model to identify and remove deep-water effects (>100 m) in SDB estimation". Weren't the NDWI and ICESat depths used to do this?

Response: The authors thank the reviewer for these beneficial comments. The authors recognize that this sentence may appear somewhat abrupt without adequate context and thus require an explanation as to why this correction is necessary. Previous studies (e.g., Jia et al., 2023) have found that in SDB modeling, it is essential to perform reference depth correction to minimize the impact of sun glint in SDB modeling. This correction helps to reduce the radiative energy contributions from the water surface and the water column, thereby improving the overall accuracy and robustness of the SDB model. First, in optical images of deep-water regions, solar radiation often causes sun glint, which interferes with the extraction of underwater energy in shallow areas, thereby affecting the accuracy of model training and depth estimation. Therefore, sun glint correction is applied to the deep-water images to stabilize image characteristics and enhance the consistency of SDB results. Second, by determining the reference radiative energy in the deep-water region and removing it from the shallow-water region, the reflection energy from the water surface and water column is eliminated, leaving only the energy reflected from the seafloor. This establishes an accurate relationship between seafloor reflection energy and water depth, and significantly improves the accuracy of the SDB.

Based on previous research Jia et al. (2023), the GEBCO model was used to select deep-water regions, and the corresponding RGB reflectance data in those regions were extracted and removed from the Sentinel-2 imagery. The deep-water correction term, represented by the parameter $\lambda_\infty$ in Eq.10 of the original manuscript, was applied to eliminate the influence of deep-water areas. Furthermore, the NDWI and ICESat-2 water depth data were utilized to construct the shallow water mask for the island reefs. For a more detailed explanation, please refer to Q7. Based on the reviewer's comments, the authors have added more explanation of the deep-water effect and its impact in the revised manuscript, making the discussion clearer and more accurate, please refer to lines 185-188.

Q5. Line 320-335. This is not clear. Dividing the track into segments "based on water depth variation trend (from ascend to descend)".  Divide how?  This is critical to  how the correlation coefficients will be determined.  The red band will disappear much sooner than green or blue.  "Ascend to descend" should be changed, they are actions so it doesn't make sense.  "Shallow to deep"?

Response: The authors thank the reviewer for these beneficial comments. The authors realize that the explanation in this section may not be sufficiently clear and could potentially cause confusion for readers. The authors' intention here is to segment the ICESat-2 depth point cloud data into specific intervals and then perform Pearson correlation analysis with the corresponding RGB reflectance information from Sentinel-2. This analysis helps to identify and remove outlier or noisy data, thereby improving the robustness and noise resistance of the SDB model.

Due to the influences of water quality, bottom conditions, and other factors on ICESat-2 point cloud data, as well as the effects of lighting and atmospheric conditions on Sentinel-2, significant anomalies may arise between the along-track water depth data from ICESat-2 and the corresponding reflectance data from Sentinel-2 pixels. Such anomalies can adversely affect the regression parameter estimation based on the LBM model. Consequently, the authors segmented the ICESat-2 depth data and correlated it with each band of Sentinel-2 reflectance data to filter out low-quality data points. A correlation threshold of $p = 0.4$ was chosen. When the correlation coefficients for two or more bands were below this threshold, the data was considered of low quality and excluded from the SDB modeling. Additionally, since the modeling region is located in shallow water, the red band reflectance does not attenuate to zero, allowing the correlation analysis and SDB modeling to still be performed. Based on the reviewer's comments, the authors have revised the description to specify that the correlation analysis was conducted with a step size of 500 m. Please refer to lines 364-368 in the revised manuscript for more information.

Q6. After that section, how is SDB determined for the whole island? H_0 and h_i were determined for each track "segment". Then what? Were they interpolated or averaged?

Response: The authors thank the reviewer for these beneficial comments. The authors realize that the explanation regarding the input data and output products for LBM modeling in the original manuscript may not have been sufficiently clear, potentially causing confusion for the readers. In the SDB modeling process based on the LBM model, the authors first define the SDB modeling region as the shallow water area around a specific island or reef. All valid ICESAT-2 water depth data and their corresponding Sentinel-2 reflectance information within the shallow water mask of that region are then used as input data for the LBM. The authors perform a multiple linear regression analysis to estimate the coefficients $h_i$, and the trained $h_i$ coefficients (i.e., the trained LBM) are then applied to the entire shallow water region's Sentinel-2 reflectance data to estimate the SDB. For a more detailed explanation of the modeling process, please refer to Q3. It is important to note that the segmentation of the ICESat-2 data and the Pearson analysis serves not to estimate the $h_i$ coefficients, but rather to identify and remove noisy data. For detailed explanation regarding data segmentation, please refer to Q5. Please refer to lines 243-244, lines 364-368 and lines 384-386 in the revised manuscript for more explanation about the SDB estimation and ICESat-2 track segmentation.

Q7. And were the correlation coefficients determined for all locations on the track within the shallow water mask. The shallow water mask was determined by the intersection of the NDWI and ICESat?. And how was ICESat screened, line 174?

Response: The authors thank the reviewer for these beneficial comments. As explained by the authors in Q3 and Q6 regarding the SDB modeling process and the input-output data, all ICESat-2 water depth data within the shallow water mask of a specific island or reef are used for SDB modeling. To enable independent validation of the SDB results, certain tracks of ICESat-2 water depth data are manually selected and divided into training and validation datasets in an 8:2 ratio.

Regarding the determination of the shallow water mask for the islands and reefs, it is true that the shallow water mask in this study is based on the intersection of NDWI and ICESat-2 data. The authors initially used Sentinel-2-derived NDWI for preliminary selection. However, there are two primary limitations when using NDWI alone to construct the shallow water mask. First, the relatively low spatial resolution of Sentinel-2 results in imprecise mask boundaries. More importantly, NDWI differentiates water bodies and land based on the reflectance differences between spectral bands. However, since this study mainly focuses on shallow water areas (with minimum depths < 10 m), the bottom variation significantly affect light scattering and absorption. This causes NDWI to perform poorly in shallow waters, leading to misclassification issues. Thanks to its high resolution and ability to capture both water surface and bottom signals, ICESat-2 data compensates for the limitations of NDWI, enabling the identification of shallow water areas in regions where NDWI is less effective or in more complex environments.

Regarding the selection of ICESat-2 data, the authors first employed the water depth point cloud extraction algorithm proposed in this study, including point cloud density analysis, along-track density accumulation, sea surface point cloud identification, and water depth point cloud extraction. In conjunction with Sentinel-2 imagery, when ICESat-2 data for a given pixel detected valid water depth point clouds, the authors classified that pixel as a shallow water area rather than land. Finally, the shallow water mask was re-checked for potential misclassification issues through visual interpretation to ensure the accuracy of the identified shallow water regions. Please refer to lines 189-196 for more information.

Q8. The split of data was "80% training and 20% validation". What does this mean? Was this random, were non-overlapping ICESat transects left out of training? If not, and a random split was used, the validation is not independent. It fails to consider spatial autocorrelation (there are a lot of papers on this topic), which would bias in favor of the results. The study "validation" does not need to be redone, but this problem needs to be clearly identified, and text calling it a validation should be changed. Perhaps saying that "Model consistency was evaluated. "

Response: The authors thank the reviewer for these beneficial comments. The authors realize that the criteria for selecting training and validation data, as well as their distribution, were not clearly explained. Regarding the independence of validation data, the authors did not randomly select training and validation data. Instead, entire tracks of ICESat-2 data were manually selected for use as training or validation data, with each track being used for only one purpose. This approach not only ensures the independence and uniform spatial distribution of the training and validation data but also allows for better data control and good spatial coverage of the validation data. Moreover, numerous studies have used this method to independently validate SDB modeling results (e.g., Ma et al, 2020; Wu et al., 2023). As noted by the reviewer in Q9, airborne laser or shipborne sonar bathymetric data can provide a better and more independent validation of SDB results. However, the authors also noted in the introduction that bathymetric data from airborne or shipborne measurements tend to be sparsely distributed due to the high costs and limited coverage of these techniques. Given that the SDB results presented in this study cover over 120 islands and reefs in the South China Sea, spanning more than 1000 km$^2$, it is not feasible to conduct a global validation using airborne or shipborne data. On the other hand, the spatial distribution of ICESat-2 tracks make it more convenient for performing global model validation (Ma et al., 2020, Wu et al., 2023).

Furthermore, since the authors conducted individual SDB modeling for each island or reef, a detailed selection of training and validation data for each island was performed, resulting in a final data split of approximately 80% for training and 20% for validation. For example, for Huaguang Reef, a total of 96 valid ICESat water depth tracks were obtained, of which 72 tracks were used for training and the remaining 24 for validation, with their distribution illustrated in Fig. 4 below. This approach not only allows for better data control but also ensures good spatial coverage of the validation data. Based on the reviewer's comments, the authors have added more explanations regarding the selection of training and validation data in the revised manuscript, please refer to lines 147-150.

[Figure]

Fig. 4 Training and validation data distribution in Huaguang Reef

Q9. Figure 12 and Table 3 do provide one independent validation, as the lidar was not used for training.

Response: The authors thank the reviewer for these beneficial comments. The authors agree with the reviewer that airborne lidar bathymetry data can provide a better and more independent validation of SDB results. However, given that the SDB results in this study encompass over 120 islands and reefs spanning more than 1000 km in the South China Sea, it is not feasible to conduct a comprehensive validation using airborne or shipborne data. On the other hand, ICESat's spatial distribution allows for more convenient global model validation (Ma et al. 2020, Wu et al., 2023). It is important to note that, as explained by the authors in Q8, in consideration of the spatial coverage of ICESat tracks, the authors manually selected entire tracks of data for training or validation purposes, with each track being used for only one purpose (either training or validation). This ensures the independence of the data, please refer to lines 147-150 in the revised manuscript for more information.

Q10. On statistisics. In spite of the popularity of Rsquared (R2) as a validation metric, it is both a poor error metric and it is redundant to RMSE (and so unnecessary). And R2 cannot be compared for samples with different ranges (variance in X). Many statisticians have reported this; King 1986

(https://www.jstor.org/stable/2111095) is a good example. There are several descriptions of the problem on the web (R2 is the fit of the line against the variance in the data, so a wider range of data will have a higher R2). Figure 9 shows the problem. Compare 9e to 9b. Occurring to R2, 9e (0.938) outperforms 9b (0.878). However, 9e has twice the error, 1.631 vs 0.802 for 9b. R2 does not provide useful information. Why? 9e has twice the range of depths, so the squared variance is much greater. It's ok to leave the R2 in the figures, because there are people who are desperate to see it, but leave any comparisons of R2 out of the text. Remove R2 reference from 362-374, 410-425, 529. This problem should be stated at line 341: e.g., "R2 is actually redundant with RMSE. However, R2 also varies with data range, so unlike RMSE, R2 values cannot be meaningfully compared between different samples. R2 values are included because they are familiar."

Response: The authors thank the reviewer for these beneficial comments. The authors fully agree with the reviewer's comments, which is a very valuable point. $R^2$ should not be used as the final criterion for accuracy but rather as an indicator for correlation analysis. Based on the reviewer's comments, the authors have removed the explanation of the $R^2$ parameter and use RMSE to evaluate the SDB modeling results, thereby improving the rigor of the manuscript. Please refer to lines 405-407 for more information. Also, based on the comment of the reviewer 2, the figures 9 and 10 are redesigned, where the point cloud data from regression analysis have been resampled to achieve a more uniform depth distribution. The statistics of these two figures are thus updated, please refer to the updated Figure 9 (line 430) and Figure 10 (line 435) and lines 416-429 for more information.

Q11. Line 335. "The effects of deep-water areas were then removed to minimize the influence of bottom reflection on SDB estimation". What effects were removed from what? Does this text belong before line 312? ("average deep-water reflectance").

Response: The authors thank the reviewer for these beneficial comments. The authors realize that this statement appears abruptly and need to explain why it is relevant. Studies by Jia et al. (2023) show that correcting for reference depth in SDB modeling mitigates sun glint effects, improving the accuracy and robustness of depth estimations from optical images. Therefore, sun glint correction is applied to the deep-water images to stabilize image characteristics and enhance the consistency of SDB results. In addition, by determining the reference radiative energy in the deep-water region and removing it from the shallow-water region, the reflection energy from the water surface and water column is eliminated, leaving only the energy reflected from the seafloor. This establishes an accurate relationship between seafloor reflection energy and water depth, and significantly improves the accuracy of the SDB.

In this study, GEBCO data was used to identify the deep-water areas, and the corresponding RGB reflectance information of the Sentinel-2 pixels in these deep-water regions was selected. This information was then removed from the Sentinel-2 images, as represented by the parameter $\lambda_\infty$ in Equation 10 (line 312), which accounts for the deep-water correction. Based on the reviewer's comments, the authors have further elaborated on the deep-water effect and its impact in the revised manuscript, aiming to make the article clearer and more precise. Please refer to lines 185-188, line 243, and lines 380-382 for more information.

Q12. Figure 12 and 13 captions are not clear, please include the letters in the caption. It would be even better to label each column.

Response: The authors thank the reviewer for these beneficial comments. Due to the large number of subfigures in Figures 12 and 13, the explanations in the caption section of the original article were too general, making the pictures less readable. Based on the reviewer's comments, more descriptions of the subfigures in these two figures were added in the caption. Please refer to page 21 Figure 12, and page 22 Figure 13 for more information.

Reference

Jia, D., Li, Y., He, X., Yang, Z., Wu, Y., Wu, T., and Xu, N.: Methods to Improve the Accuracy and Robustness of Satellite-Derived Bathymetry through Processing of Optically Deep Waters, Remote Sens., 15, 5406, https://doi.org/10.3390/rs15225406, 2023.

Wu, Y., Li, Y., Jia, D., Andersen, O. B., Abulaitijiang, A., Luo, Z., and He, X.: Seamless seafloor topography determination from shallow to deep waters over island areas using airborne gravimetry, IEEE Trans. Geosci. Remote Sens., 61, 1–19, https://doi.org/10.1109/TGRS.2023.3336747, 2023.

Ma, Y., Xu, N., Liu, Z., Yang, B., Yang, F., Wang, X. H., and Li, S.: Satellite-derived bathymetry using the ICESat-2 lidar and Sentinel-2 imagery datasets, Remote Sens. Environ., 250, 112047, https://doi.org/10.1016/j.rse.2020.112047, 2020.

---

## Author Comment (AC2)

This manuscript integrated ICESat-2 data and multispectral imagery from Sentinel-2 to construct a high-resolution, high-accuracy shallow water depth model namely HHU24SWDSCS. In geopolitically sensitive areas, such as the South China Sea, where the seafloor topography is complex, existing water depth data primarily rely on sparse multibeam sounding technology and satellite altimetry-derived depths. The HHU24SWDSCS model developed in this study successfully filled the gap in shallow water areas and offered an alternative that can be applied to similar regions globally. The comparisons with existing bathymetry models demonstrate that the computed model offers significant advantages in both accuracy and details for shallow areas, highlighting its potential for high-quality shallow water depth measurement. The manuscript demonstrates notable innovation and scientific significance. Publication is recommended following revisions.

Q1: The English writing should be further polished.

Response: The authors thank the reviewer for these beneficial comments. The authors have thoroughly reviewed the entire manuscript and made revisions to address grammatical errors and improve language expressions, resulting in more accurate and fluent expressions. The authors believe these improvements have significantly enhanced the readability and clarity of the work. Please refer to the revised manuscript.

Q2: Line 19: What is the full name of HHU24SWDSCS?

Response: The authors thank the reviewer for these beneficial comments. The full name of the shallow-water bathymetry model developed in this study is 'Shallow-Water Depth (SWD) Model for the South China Sea (SCS) developed by Hohai University in 2024,' which has been abbreviated as 'HHU24SWDSCS' (Hohai University 2024 Shallow-Water Depth Model of South China Sea). This nomenclature reflects the model's specific application to shallow-water bathymetry in the South China Sea, its development institution, and the year of creation. Based on the comment of the reviewer, the authors have added a detailed explanation of the model name in the revised manuscript. Please refer to lines 19-20 and lines 95-96 for more information.

Q3: Line 31: How to define the shallow water in the study?

Response: The authors thank the reviewer for these beneficial comments. This study focuses on Satellite-Derived Bathymetry (SDB) modeling in shallow waters around the South China Sea (SCS) islands, primarily based on two considerations: (1) these areas are challenging for shipborne or airborne bathymetric surveys due to political and cost constraints, and (2) the relatively better water quality surrounding these islands, which provides favorable conditions for high-precision bathymetry modeling. However, the prior water depth information is constrained by the penetration capability of ICESAT-2 and the reflectance attenuation characteristics of Sentinel-2. In the clear waters of the South China Sea, ICESAT-2 can achieve bathymetric measurements up to ~30 m. Furthermore, existing research indicates that the mapping function between ICESAT-2 depth information and Sentinel-2 reflectance maintains a favorable linear relationship within the 30-40 m depth range, which significantly enhances the accuracy of SDB. Therefore, the study defines the shallow water zone as the intersection of effective detection depths from both datasets, subsequently

limiting the study area to nearshore waters within 30-40 m depth. Based on the comment of the reviewer, the authors have supplemented and refined the detailed description of the shallow water zone definition in the revised manuscript. Please refer to lines 124-126 for more information.

Regarding the determination of the shallow water mask for the islands and reefs, the shallow water mask in this study is based on the intersection of NDWI and ICESat-2 data. The authors initially used Sentinel-2-derived NDWI for preliminary selection. However, there are two primary limitations when using NDWI alone to construct the shallow water mask. First, the relatively low spatial resolution of Sentinel-2 results in imprecise mask boundaries. More importantly, NDWI differentiates water bodies and land based on the reflectance differences between spectral bands. However, since this study mainly focuses on shallow water areas (with minimum depths < 10 m), the bottom variation significantly affect light scattering and absorption. This causes NDWI to perform poorly in shallow waters, leading to misclassification issues. Thanks to its high resolution and ability to capture both water surface and bottom signals, ICESat-2 data compensates for the limitations of NDWI, enabling the identification of shallow water areas in regions where NDWI is less effective or in more complex environments. Based on the comment of the reviewer, the authors have added a detailed explanation of the shallow water mask extraction. Please refer to lines 189-196 for more information.

Q4: Line 105: It is seen that the authors only retrieved water depths over island areas, is it possible to perform SDB modeling in nearshore areas (e.g., estuarine region), and what is the SDB quality there?

Response: The authors thank the reviewer for these beneficial comments. Indeed, while SDB technology demonstrates feasibility for bathymetric mapping in nearshore areas (e.g., estuarine region), it presents significant technical challenges. The accuracy of SDB modeling in estuarine region is constrained by multiple factors, including the penetration depth limitations of ICESAT-2, reflectance attenuation characteristics of Sentinel-2, and water quality parameters such as chlorophyll concentration, bottom type, and turbidity. These constraints inevitably lead to reduced data quality and availability. In addition, previous studies, such as Xu et al. (2022), have successfully implemented SDB modeling in inter-tidal areas, demonstrating its potential for investigating sediment dynamics. However, this study has excluded these regions from its scope considering the complex influences of nearshore water quality and seafloor variations on SDB modeling accuracy. Based on the comment of the reviewer, the authors have expanded the research outlook in the conclusion section, highlighting the potential for future in-depth investigations into SDB applications in estuarine environments. Please refer to lines 594-596 for more information.

Q5: Table 1: More information should be further shown, like max depth, min depth, mean depth, etc.
Table 1 and Table 2: These tables should follow the three-line table format.

Response: The authors thank the reviewer for these beneficial comments. Based on the comment of the reviewer, the table formatting issues have been corrected in the revised manuscript. Additionally, the SDB depth ranges for each region have been described in the context of Figure 8. Please refer

to lines 402-411 for more information.

Q6: Line 145: The explanation regarding the selection of training and validation data is not sufficiently clear. Why was an 8:2 rule used for training and validation? How about the data distribution?

Response: The authors thank the reviewer for these beneficial comments. The authors realize that the criteria for selecting training and validation data, as well as their distribution, were not clearly explained. In this study, the entire tracks of ICESat-2 data were manually selected for use as training or validation data, with each track being used for only one purpose. This approach not only ensures the independence and uniform spatial distribution of the training and validation data but also allows for better data control and good spatial coverage of the validation data. Moreover, numerous studies have used this method to independently validate SDB modeling results (e.g., Ma et al, 2020; Wu et al., 2023). Although airborne laser or shipborne sonar bathymetric data can provide a better and more independent validation of SDB results, the authors also noted in the introduction that bathymetric data from airborne or shipborne measurements tend to be sparsely distributed due to the high costs and limited coverage of these techniques. Given that the SDB results presented in this study cover over 120 islands and reefs in the South China Sea, spanning more than 1000 km$^2$, it is not feasible to conduct a global validation using airborne or shipborne data. On the other hand, the spatial distribution of ICESat-2 tracks makes it more convenient for performing global model validation.

Furthermore, since the authors conducted individual SDB modeling for each island or reef, a detailed selection of training and validation data for each island was performed, resulting in a final data split of approximately 80% for training and 20% for validation. For example, for Huaguang Reef, a total of 96 valid ICESat water depth tracks were obtained, of which 72 tracks were used for training and the remaining 24 for validation, with their distribution illustrated in Fig. 1 below. Based on the reviewer's comments, the authors have added more explanations regarding the selection of training and validation data in the revised manuscript, please refer to page 5 line 147-150.

[Figure]

Fig. 1 Training and validation data distribution in Huaguang Reef

Q7: Figure 3: How did the authors perform water mask in the imagery? Since the study focuses on

shallow water areas, it is easy to cause confusion between land and water in Sentinel-2 imagery.

Response: The authors thank the reviewer for these beneficial comments. Regarding the determination of the shallow water mask for the islands and reefs, the shallow water mask in this study is based on the intersection of NDWI and ICESat-2 data. The authors initially used Sentinel-2-derived NDWI for preliminary selection. However, there are two primary limitations to using NDWI alone to construct the shallow water mask. First, the relatively low spatial resolution of Sentinel-2 results in imprecise mask boundaries. More importantly, NDWI differentiates water bodies and land based on the reflectance differences between spectral bands. However, since this study mainly focuses on shallow water areas (with minimum depths < 10 m), the bottom significantly affect light scattering and absorption. This causes NDWI to perform poorly in shallow waters, leading to misclassification issues. Thanks to its high resolution and ability to capture both water surface and bottom signals, ICESat-2 data compensates for the limitations of NDWI, enabling the identification of shallow water areas in regions where NDWI is less effective or in more complex environments. Based on the comment of the reviewer, the authors have added a detailed explanation of the shallow water mask extraction. Please refer to lines 189-196 for more information.

Q8: Section 3.1: What is the bounding depth detected by IceSat-2? What factors can affect the ability of IceSat-2-based depth detection?

Response: The authors thank the reviewer for these beneficial comments. In the study area (i.e., SCS), the water clarity enables ICESat-2 to effectively penetrate water depths of approximately 30 m. Its 532 nm laser wavelength demonstrates good penetration capability in clear waters; however, the laser signal undergoes attenuation due to the optical properties of water during propagation, leading to measurement failure in deeper regions beyond 30 m.

More importantly, several environmental factors, such as water clarity, wave conditions, suspended matter concentration, and seabed reflectivity characteristics, also limit the detection depth range. Seabed characteristics play a crucial role in measurement accuracy. The flat sandy or rocky seabed can produce strong return signals, thereby enhancing measurement precision, while soft sediments (e.g., muddy bottoms) or complex topographies tend to cause laser signal scattering or absorption, resulting in weaker return signals and reduced measurement accuracy. Based on the comment of the reviewer, more explanation about the bounding depth of ICESAT-2 is added in the revised manuscript, please refer to lines 145-146 for more information.

Q9: Section 3.2: What is the bounding depth detected by Sentinel-2? The description of the methodology may be improved. For instance, the choice of modeling region and the reasons for data segmentation and Pearson correlation analysis are not adequately explained.

Response: The authors thank the reviewer for these beneficial comments. This study primarily utilizes the red, green, and blue (RGB) bands of Sentinel-2 multispectral imagery for SDB modeling. The detection depth range of these bands is influenced by factors such as water transparency, band-specific attenuation characteristics, and seabed reflectance properties. Firstly, water transparency is a critical factor determining the depth-detection capability of the RGB bands. In clear water, light

signals can penetrate deeper layers. However, in turbid waters, suspended particles (e.g., sediments, plankton) significantly increase light scattering and absorption, leading to rapid signal attenuation and limiting detection depth. Secondly, the attenuation rates of different bands vary. The red band attenuates the fastest and has the weakest penetration capability, while the blue and green bands exhibit stronger underwater detection capabilities. Additionally, the reflectance properties of the seabed significantly affect detection accuracy. Hard seafloor (e.g., sandy or rocky bottoms) reflect stronger signals than the soft sedimentary bottoms or complex seabed terrains. Based on the SDB modeling results for the SCS presented in this study, Sentinel-2 can achieve reasonable seabed topography delineation within a depth range of 30-40 m. This indicates that its effective detection depth in clear waters can reach up to 30-40 m.

In this study, the shallow water areas surrounding the islands and reefs were chosen as the SDB modeling regions, as shown in Figs. 1 and 8 in the original manuscript. Due to limitations in image quality and water quality conditions, the authors restricted each SDB modeling region to either a single island or a group of nearby reefs. For example, in Fig. 3(a) and Table 2 of the original manuscript, typical island regions in the Xisha and Nansha areas are presented, along with the images and ICESat-2 data used for SDB modeling in each region. Taking the Yongle Atoll as an example (Fig. 2 below), it includes several small islands such as Ganquan Island and Jinyin Island. Due to the relatively concentrated spatial distribution of these islands, the authors assumed that the Sentinel-2 image quality, lighting conditions, and other factors would be similar across these regions. As such, a unified regression model was applied to the entire group of islands in the region.

The authors realize that the explanation for the Pearson analysis may not be sufficiently clear and could potentially cause confusion for readers. In fact, the authors' intention here is to segment the ICESat-2 depth point cloud data into specific intervals and then perform Pearson correlation analysis with the corresponding RGB reflectance information from Sentinel-2. This analysis helps to identify and remove outlier or noisy data, thereby improving the robustness and noise resistance of the SDB model. Due to the influences of water quality, bottom conditions, and other factors on ICESat-2 point cloud data, as well as the effects of lighting and atmospheric conditions on Sentinel-2, significant anomalies may arise between the along-track water depth data from ICESat-2 and the corresponding reflectance data from Sentinel-2 pixels. Such anomalies can adversely affect the regression parameter estimation based on the linear band model (LBM) model. Consequently, the authors segmented the ICESat-2 depth data and correlated it with Sentinel-2 reflectance data to filter out low-quality data points. A correlation threshold of $p = 0.4$ was chosen. When the correlation coefficients for two or more bands were below this threshold, the data was considered of low quality and excluded from the SDB modeling. Based on the comments of reviewer 1, the authors have revised the description to specify that the correlation analysis was conducted with a step size of 500 m. Based on the comment of the reviewer, more explanation about the method and choice of modeling region is added in the revised manuscript, please refer to lines 173-178, lines 189-196, and lines 364-367 for more information.

[Figure]

Fig. 2 the Sentinel-2 imagery (background), ICESat-2 water depth (tracks), shallow water mask (white polygons) of the Yongle island region

Q10: Line 310: How generalizable is the LBM model trained by the authors? Can it be applied to other marine areas with insufficient ICESat-2 data for SDB modeling?

Response: The authors thank the reviewer for these beneficial comments. Indeed, the accuracy of SDB modeling based on ICESat-2 data and Sentinel-2 imagery is significantly influenced by factors such as water quality, seabed characteristics, and solar illumination in shallow waters. This is the primary reason why the SDB modeling regions have been restricted to individual islands and their adjacent areas in this study. Similarly, the generalization performance of the LBM is constrained by several factors, including the representativeness and quantity of the modeling data, as well as the quality of Sentinel-2 imagery. In general, LBM is a regression model, and its generalization performance improves significantly when more representative and higher-quality data are used for training. However, due to the limited parameter number of the LBM and the lack of consideration for environmental factors such as seabed characteristics and turbidity, as well as the absence of physical constraints, the modeling performance of LBM can vary greatly across different regions. To further enhance the model's generalization performance and enable its application in areas with sparse ICESat-2 data, the author suggests improvements in two key aspects: first, selecting a larger and more representative dataset (covering various depths, times, turbidity levels, and seabed types); second, incorporating machine learning or deep learning frameworks with physical constraints in SDB modeling. This would enable a more comprehensive consideration of influencing factors, optimize model parameter training, and ultimately improve the robustness and transferability of SDB modeling. The author sincerely appreciates the reviewer's valuable suggestions, which have provided new directions for future research. Based on the reviewer's comments, the author has added more insights on the generalization capability of the model in the conclusion section and will explore this issue further in subsequent studies. Please refer to lines 589-591 for more information.

Q11: Line 350: How deep of water depths can be detected from SDB (seems ~ 30 m in this study area), is it possible to apply SDB modeling over water areas with deeper depths than the one used in this study?

Response: The authors thank the reviewer for these beneficial comments. The SDB technique is limited by its inability to detect seafloor topography in deep waters, where the SDB over the water

areas with depths deeper than 30 m may become unreliable. This impedes the extensive applications of the SDB technique, and the acquirement of water depths over deep water areas largely relies on in situ data acquired from echo soundings, LiDAR data, and seismic depths. Please refer to L402-411 and lines 556-557 for more information.

Q12: Figure 8: The authors mentioned several islands and reefs in their model but omit other islands and reefs in the South China Sea, such as the ones in the Zhongsha Islands region. Please explain this issue.

Response: The authors thank the reviewer for these beneficial comments. The authors have excluded the Zhongsha Islands from the SDB modeling scope primarily due to the challenges in acquiring high-quality ICESat-2 bathymetric data in this region. The detection capability of ICESat-2 is influenced by multiple factors, including water transparency, suspended sediment concentration, and seabed reflectivity characteristics. Although the Zhongsha Islands area maintains relatively clear waters, the sandy seabed and greater water depth compared to the coral reef regions of the Xisha and Nansha Islands present additional challenges. Furthermore, the geological characteristics of the seabed may cause scattering or absorption of the laser signal during transmission, resulting in weakened return signals and reduced data availability. Within the Zhongsha Islands region, only the Huangyan Island area has yielded relatively reliable ICESat-2 bathymetric data, while data availability remains poor in other areas such as the large atoll regions. As a result, the authors have excluded this region from the current study. Based on the comment of the reviewer, the authors have provided additional explanatory details in the revised manuscript to clarify this issue. Please refer to lines 119-121 for more information.

Q13: Figures 9 and 10: The point cloud density was uneven, with most points concentrated in the 1-3 meter depth range. While Figure 7 shows that the ICESat-2 data was predominantly concentrated in shallow water areas, this could lead to inconsistent fitting of the regression model across different depth ranges. I suggest the authors consider resampling the ICESat-2 data within the 1-3 meter range.

Response: The authors thank the reviewer for these beneficial comments. The authors acknowledge the formatting inconsistencies in the regression analysis results presented in Figures 9 and 10. Based on the comment of the reviewer, the point cloud data from regression analysis have been resampled to achieve a more uniform depth distribution. Also, the statistics of these two figures are updated. Please refer to revised Figure 9 and 10.

Q14: Figure 11: Similar to the previous figures, Figure 11 showed uneven point cloud density distribution. Additionally, the origin of the XY axes should be at 0 m, rather than -10 m. Please redraw these figures.

Response: The authors thank the reviewer for these beneficial comments. The authors have thoroughly revised the modeling dataset by implementing depth-stratified resampling to achieve a uniform data distribution, thereby enhancing the reliability and validity of the regression analysis results. Based on the comment of the reviewer, Figure 11 has been completely redesigned, with

particular attention paid to optimizing the axis scaling and presentation of the regression analysis plots. Please refer to revised Figure 11.

Q15: Line 440: There was no introduction of the DTU18BAT, topo_27.1, GEBCO_2023, or SRTM models earlier in the manuscript. What data these models use for construction? What are their spatial resolutions and accuracies? More information can be included.

Response: The authors thank the reviewer for these beneficial comments. In this study, several recently released global bathymetry models are considered, including DTU18BAT, topo_25.1, SRTM15+V2.5.5, and GEBCO_2023. DTU18BAT (1′×1′) is a latest version of series of models developed by the Technical University of Denmark (DTU) in 2019 (https://ftp.space.dtu.dk/pub/DTU18/1_MIN/). DTU18BAT was constructed by a combination of the GEBCO model and satellite altimetric gravity anomalies; and this model included 3 years of Sentinel-3A and 7 years of Cryosat-2 data and used FES2014 for ocean tide correction. The topo_25.1 (1′×1′) is a recently released model developed by the Scripps Institution of Oceanography (SIO) (https://topex.ucsd.edu/pub/global_topo_1min/), and it was predicted from ship soundings and satellite altimetric gravity anomalies. SIO has developed several versions of bathymetry models, and the models have been consistently improved in terms of accuracy and resolution with the accumulation of satellite altimetry data. SRTM15+V2.5.5 (15″×15″) is an updated version of SRTM15+ series of models that were developed in SIO (https://topex.ucsd.edu/pub/srtm15_plus/). These models were computed by combining shipboard soundings and depths predicted from satellite altimetric gravity data. GEBCO_2023 was a recently released model developed by the GEBCO Bathymetric Compilation Group in 2023 (https://www.gebco.net/data_and_products/gridded_bathymetry_data/). GEBCO_2023 was developed by using SRTM15+V2.5.5 as the base data, and included in-situ depths, such as echo soundings, seismic records, and lidar data. GEBCO_2023 was well constrained where in-situ data were available, whereas interpolation was implemented from satellite altimetry-predicted depths and bathymetric soundings at locations where in-situ records were missing. Based on the comment of the reviewer, the authors have provided additional introduction in the revised manuscript, please refer to lines 220-237 for more information.

Q16: Line 495: The authors have used ICESat-2 data spanning over five years for SDB modeling. Have they considered the potential impact of temporal changes in seafloor topography due to human activities or ocean currents during this period?

Response: The authors thank the reviewer for these beneficial comments. The local bathymetry is affected by the temporal changes caused by sediment deposition, erosion, ocean current, human activities and etc. The human induced influence is neglected, due to fact that the study area is a remote region and the effect of human activities here is relatively weak. However, other temporal factors may have potential influences on underwater topography. According to previous studies, we assume that the temporal changes of underwater topography have minor influences on water depth estimation over short periods in the study area. Moreover, we choose the Sentinel-2 images that fall within the time spans of the ICESat-2 data, which are close to the overpass time of the ICESat-2 data (from 2018 to 2023) used in this study. In such a way, the effects of the temporal changes on

local bathymetry estimation can be largely reduced. Please refer to lines 169-170 for more information.

Q17: Line 520: The conclusion contained some repetitive expressions and redundant language. It can be streamlined for conciseness.

Response: The authors thank the reviewer for these beneficial comments. The conclusion section has been substantially revised to enhance its precision and focus. The authors have carefully reviewed the entire manuscript to correct grammatical errors and improve language clarity. These revisions have significantly improved the overall quality and readability of the manuscript. Please refer to lines 572-596 for more information.

Q18: Line 545: The appendix lists all the islands and reefs used for modeling, but does the author model each of these individually? How much data was used for each island or reef?

Response: The authors thank the reviewer for these beneficial comments. In this study, the shallow water areas surrounding the islands and reefs were chosen as the SDB modeling regions, as shown in Figs. 1 and 8 in the original manuscript. Due to limitations in image quality and water quality conditions, the authors restricted each SDB modeling region to either a single island or a group of nearby reefs. For example, in Fig. 3(a) and Table 2 of the original manuscript, typical island regions in the Xisha and Nansha areas are presented, along with the images and ICESat-2 data used for SDB modeling in each region. Taking the Yongle Atoll as an example (Fig. 3 below), it includes several small islands such as Ganquan Island and Jinyin Island. Due to the relatively concentrated spatial distribution of these islands, the authors assumed that the Sentinel-2 image quality, lighting conditions, and other factors would be similar across these regions. As such, a unified regression model was applied to the entire group of islands in the region. The authors first selected the high-quality Sentinel-2 L2A image (20210817T025549) and extracted the RGB band information within the shallow water mask region. Additionally, as confirmed in the study by Jia et al. (2023), applying deep-water reflectance correction to Sentinel-2 imagery significantly improves the accuracy of SDB modeling. Based on the GBCO empirical model, the authors identified regions with water depths exceeding 100 m in the modeling area. The RGB reflectance of these deep-water regions was extracted and used as correction information to eliminate sun glint and other surface effects in the RGB data.

[Figure]

Fig. 3 the Sentinel-2 imagery (background), ICESat-2 water depth (tracks), shallow water mask (white polygons) of the Yongle island region

Next, the authors used 19 ICESat-2 tracks containing sea floor information as prior water depth data, which underwent processing steps including water depth point cloud extraction, noise removal, refraction correction, and reference datum unification. Due to the presence of significant Gaussian noise in the depth point cloud, the authors applied piecewise linear fitting to identify the mid-line of the water depth point cloud, which was then used as the true water depth. The depth range was subsequently gridded, and the fitted water depth values were uniformly sampled within grids of different depths, ensuring an equal distribution of point cloud data across varying depth levels. Additionally, as the ICESat-2 data has a significantly higher along-track resolution (0.7 m) compared to the Sentinel-2 spatial resolution (10 m), the authors computed the average depth for the ICESat points located within each pixel and used this averaged value as the prior water depth for that pixel. This process ensures that the prior depth data is consistent with the spatial resolution of the Sentinel-2 imagery.

Finally, the authors extracted the RGB reflectance information from each Sentinel-2 pixel along the ICESat-2 track and performed multiple linear regression fitting using the LBM model, relating this reflectance data to the ICESat-2 prior water depth. Through this process, the regression parameters were estimated. The trained model was then applied to the Sentinel-2 reflectance data, resulting in the SDB model for the shallow water region of the Yongle island.

It is important to note that the islands and reefs mentioned in line 170 of the original manuscript were selected as typical examples from the Xisha and Nansha regions to facilitate the demonstration of the SDB modeling process and its details. The regression modeling areas for these islands are located within the shallow water mask regions shown in Fig. 3 of the original manuscript (inside the white shapes). For all other islands, the corresponding regression modeling areas can be found in the shallow water regions of Fig. 1 of the original manuscript (inside the gray shapes). Additionally, the content in lines 320-325 focuses on selecting the ICESat-2 water depth data and Sentinel-2 RGB reflectance information with the highest correlation based on Pearson correlation analysis. This step is critical to avoid including outliers or noise that could affect the SDB modeling. Based on the reviewer's comments, the authors have further clarified the issue of regression modeling areas in the revised manuscript and have expanded the explanations related to data selection and the modeling approach, making the manuscript clear and comprehensive, please refer to lines 173-177, line 189-196, Figure 3 and Table 2 for more information.

Q18: The manuscript contains several expressions like "point clouds," where "point" should be replaced with "data".
References: The format of literatures should be further normalized.

Response: The authors thank the reviewer for these beneficial comments. The authors have carefully revised and corrected this expression throughout the entire manuscript. Additionally, the authors have conducted a thorough review of the manuscript to ensure all statements are precise and scientifically accurate. Please refer to the revised manuscript.

**Reference**

Jia, D., Li, Y., He, X., Yang, Z., Wu, Y., Wu, T., and Xu, N.: Methods to improve the accuracy and robustness of satellite-derived bathymetry through processing of optically deep waters, Remote Sensing, 15, 5406, https://doi.org/10.3390/rs15225406, 2023.

Ma, Y., Xu, N., Liu, Z., Yang, B., Yang, F., Wang, X. H., and Li, S.: Satellite-derived bathymetry using the ICESat-2 lidar and sentinel-2 imagery datasets, Remote Sens. Environ., 250, 112047, https://doi.org/10.1016/j.rse.2020.112047, 2020.

Xu, N., Ma, Y., Yang, J., Wang, X. H., Wang, Y., and Xu, R.: Deriving Tidal Flat Topography Using ICESat-2 Laser Altimetry and Sentinel-2 Imagery, Geophys. Res. Lett., 49, e2021GL096813, https://doi.org/10.1029/2021GL096813, 2022.

Wu, Y., Li, Y., Jia, D., Andersen, O. B., Abulaitijiang, A., Luo, Z., and He, X.: Seamless seafloor topography determination from shallow to deep waters over island areas using airborne gravimetry, IEEE Trans. Geosci. Remote Sens., 61, 1–19, https://doi.org/10.1109/TGRS.2023.3336747, 2023.

---

## Author Response (AR2)

Reviewer 1:

The study derives a crucial shallow-water depth model over island areas in South China Sea (SCS) using the satellite-derived bathymetry technique. The results demonstrate that the computed model has significant improvements compared to traditional models, providing a reference for mapping shallow water depths close to islands over SCS. The article is well-prepared, the theoretical framework and numerical experiments are well presented. Some minor revisions are as follows:

1. P3, L105: GEBCO_2024 was recently released, the authors may consider to replace GEBCO_2023 by GEBCO_2024 for numerical experiments.

Response: The authors thank the reviewer for these beneficial comments. The authors conducted a reassessment of the latest GEBCO_2024 model. Overall, the performance of coastal shallow water bathymetry in GEBCO_2024 shows little improvement over GEBCO_2023 in the South China Sea (SCS) area, despite significant differences between the two models (Figure 1 below, the differences can exceed 100 m). This may be due to the lack of incorporation of high-precision shallow water bathymetric data in GEBCO_2024. Specifically, validation results in Lingyang Reef indicate that the SDB model demonstrates better consistency with airborne LiDAR data compared to GEBCO_2024, with RMSE values of 1.10 m for satellite-derived-bathymetry (SDB) and 32.38 m for GEBCO_2024. Furthermore, comparisons across six typical reef areas revealed that GEBCO_2024 still exhibits relatively low modeling accuracy in the shallow waters of the SCS. Based on the comments of the reviewer, the authors have replaced the GEBCO_2023 model with GEBCO_2024 in the revised manuscript. Please refer to Pages 21 to 25 (Figures 12 to 14 and Tables 3 to 5) for more information.

[Figure]

Figure 1 Differences between GEBCO_2024 and GEBCO_2023

2. P6, L160: The intelligibility of Figure 2 should be improved since the distribution of ICESat-2 data are not clearly observed over some islands.

Response: The authors thank the reviewer for these beneficial comments. The authors apologize for the reduced image clarity due to PDF processing. Based on the comments of the reviewer, the authors have re-adjusted the image quality to enhance clarity. Additionally, the authors have added the subplots of Figure 2 to the Supplementary Material to provide a clearer presentation of the distribution of ICESAT-2 data (both training and validation datasets) utilized in the SDB modeling process. Please refer to the Supplementary Material part for more information.

3. P7, L190: Why you needed to remove deep-water effects in SDB modeling? The authors may include the possible reasons.

Response: The authors thank the reviewer for these beneficial comments. According to the study by Jia et al. (2023), correcting for deep-water effects in SDB modeling helps mitigate sun glint effects, which in turn improves the accuracy and robustness of depth estimations derived from optical images. In addition, by determining the reference radiative energy in the deep-water region and removing it from the shallow-water region, the reflection energy from the water surface and water column is eliminated, leaving only the energy reflected from the seafloor. This establishes an accurate relationship between seafloor reflection energy and water depth, and significantly improves the accuracy of the SDB. Based on the reviewer's comments, the authors added more explanation about the reason of removing deep-water effects. Please refer to Page 7 Line 191 for more information.

4. P8, L200: Figure caption of Figure 3. The text "and the cyan dotted boxes in (b) and (c) indicate typical nighttime and daytime ICESat-2 tracks, respectively" can be revised as "and the cyan dotted boxes in (b) and (c) indicate the typical nighttime and daytime ICESat-2 tracks shown in Figures 6 and 7, respectively" to make it clear.

Response: The authors thank the reviewer for these beneficial comments. Based on the comment of the reviewer, the authors have refined this statement in the revised manuscript. Please refer to Page 8, Line 205 for the updated version.

5. P11, L305: "where, S1 and…" should be revised as "where S1 and…".
P15, L355 and 360: what is 'n' meaning in Eqs. (10) and (11), every variable in these equations should be predefined.

Response: The authors thank the reviewer for these beneficial comments. Based on the comment of the reviewer, the authors have carefully revised the imprecise statements. Additionally, in Eq. 10 and 11, $n$ represents the number of spectral bands used for regression modeling. Specifically, SDB modeling in this study is performed using three bands (red, green, and blue). The authors have corrected this issue in the revised manuscript and thoroughly reviewed the parameter definitions in other equations. Please refer to Page 15, Line 362 for more information.

Reference

Jia, D., Li, Y., He, X., Yang, Z., Wu, Y., Wu, T., and Xu, N.: Methods to Improve the Accuracy and Robustness of Satellite-Derived Bathymetry through Processing of Optically Deep Waters, Remote Sens., 15, 5406, https://doi.org/10.3390/rs15225406, 2023.

Reviewer 2:

The authors have revised the manuscript based on the previous comments and proposals. Some comment and proposals are following.

[1] The method used should be present in section Abstract.

Response: The authors thank the reviewer for these beneficial comments. In this study, the authors utilized a linear band model (LBM) for satellite-derive-bathymetry (SDB) modeling. Based on the comment of the reviewer, the abstract has been updated to include details about the modeling approach, please refer to Page 1 Line 21 of the revised abstract.

[2] How to unify the depth datum of all models of all islands in the study?

Response: The authors thank the reviewer for these beneficial comments. Indeed, the vertical datum of the raw ICESAT-2 photon data provided by NASA were unified to the WGS-84 ellipsoid. In this study, the ICESAT-2 photon data were processed to provide prior water depth information for SDB modeling, and were adjusted to the DTU21MSS through refraction correction and depth calculation (Eq. 9 of the manuscript). As a result, all SDB results within the study area were referenced to the DTU21MSS datum (Wu et al., 2023). For more detailed explanations regarding the vertical datum, please refer to Page 12 Line 315 and Eq. 9 of the revised manuscript.

[3] In situ depth data are commonly used to validate the depth models. But there is no in situ depth data around some islands.

Response: The authors thank the reviewer for these beneficial comments. The authors agree with the reviewer that in situ depth data (e.g., multibeam sounding data) can provide a better and more independent validation of SDB results. However, given that the SDB results in this study encompass over 120 islands and reefs spanning more than 1000 km in the South China Sea (SCS), and that the sounding data in this region are available only for waters deeper than 500 m, it is not feasible to conduct a comprehensive validation using in situ sounding data. Therefore, the authors acquired airborne LiDAR data in Lingyang Reef to perform an accuracy assessment in a localized area. Meanwhile, the distribution of ICESAT-2 data allows for more convenient global model validation (Ma et al. 2020, Wu et al., 2023). It is important to note that, in consideration of the spatial coverage of ICESAT-2 tracks, the authors manually selected entire tracks of data for training or validation purposes, with each track being used for only one purpose (either training or validation). This ensures the independence of the data. Based on the comment of the reviewer, more explanation about the validation data is added in the revised manuscript, please refer to Page 5 Line 150 for more information.

[4] How about the dependence of the method used on the sea water quality and water color?

Response: The authors thank the reviewer for these beneficial comments. The LBM used in SDB is indeed influenced by water quality and water color. Specifically, water quality factors such as turbidity and clarity affect light penetration depth, which in turn influences the usability and quality of ICESAT-2 water depth data. Similarly, water color, influenced by factors like chlorophyll-a

concentration and suspended particulate matter, alters the optical properties of water, influencing the relationship between Sentinel-2's RGB bands and ICESAT-2-derived water depths. However, in SCS region, the water quality is generally better, with superior clarity compared to areas such as river mouths. Thus, the LBM can achieve reasonable SDB results in this region. Future research plans to explore more comprehensive algorithms to incorporate multi-parameters in model training, which aims to improve the accuracy and robustness of SDB modeling by better accounting for the complex interactions between water quality, water color, and topographic features. Based on the comments of the reviewer, additional information has been added to the discussion and conclusions sections. Please refer to lines Page 23 Line 554 and Page 26 Line 595 in the revised manuscript for more details.

Reference

Wu, Y., Li, Y., Jia, D., Andersen, O. B., Abulaitijiang, A., Luo, Z., and He, X.: Seamless seafloor topography determination from shallow to deep waters over island areas using airborne gravimetry, IEEE Trans. Geosci. Remote Sens., 61, 1–19, https://doi.org/10.1109/TGRS.2023.3336747, 2023.

Ma, Y., Xu, N., Liu, Z., Yang, B., Yang, F., Wang, X. H., and Li, S.: Satellite-derived bathymetry using the ICESat-2 lidar and Sentinel-2 imagery datasets, Remote Sens. Environ., 250, 112047, https://doi.org/10.1016/j.rse.2020.112047, 2020.

---

## Author Response (AR3)

1. With the next revision, please re-check the supplement materials whether they include copies of the manuscript figures. If it is so, please remove these duplicates from the supplement archive.

In the updated supplementary material, the authors have removed the duplicated Figure 2. To ensure better correspondence between the subfigures and the original Figure 2 in the manuscript, each subfigure has been renamed with its corresponding identifier, such as "Figure 2(a).jpg".

2. Manuscript revision

The authors have thoroughly reviewed the entire manuscript, including the author information, affiliations, and the main text. An mistake was identified in the description of the GEBCO_2024 resolution, which was incorrectly stated as 15' × 15' instead of the correct 15" × 15". This mistake has been corrected in both the uploaded .doc and .pdf versions of the manuscript. Please refer to Page 3, line 26 of the revised manuscript.

3. Financial support

The authors have added two funding acknowledgments in the updated manuscript, including the project of China Railway Corporation (Grant Nos. 2021-key-14 and 2021-major-08), as well as the joint planning initiative of technology and water conservancy of Jiangxi Province (Grant No. 2022KSG01009). Please refer to the Financial Support section of the revised manuscript.